 

# Genetic circuit characterization and debugging using RNA-seq

Thomas E Gorochowski[1,†] , Amin Espah Borujeni[1,†] , Yongjin Park[1,†], Alec AK Nielsen[1], Jing Zhang[1], Bryan S Der[1], D Benjamin Gordon[1,2] & Christopher A Voigt[1,2,*]

## Abstract

Genetic circuits implement computational operations within a cell. Debugging them is difficult because their function is defined by multiple states (e.g., combinations of inputs) that vary in time. Here, we develop RNA-seq methods that enable the simultaneous measurement of: (i) the states of internal gates, (ii) part performance (promoters, insulators, terminators), and (iii) impact on host gene expression. This is applied to a three-input one-output circuit consisting of three sensors, five NOR/NOT gates, and 46 genetic parts. Transcription profiles are obtained for all eight combinations of inputs, from which biophysical models can extract part activities and the response functions of sensors and gates. Various unexpected failure modes are identified, including cryptic antisense promoters, terminator failure, and a sensor malfunction due to media-induced changes in host gene expression. This can guide the selection of new parts to fix these problems, which we demonstrate by using a bidirectional terminator to disrupt observed antisense transcription. This work introduces RNA-seq as a powerful method for circuit characterization and debugging that overcomes the limitations of fluorescent reporters and scales to large systems composed of many parts.

**Keywords** biofab; combinatorial logic; omics; synthetic biology; systems biology

**Subject Categories** Genome-Scale & Integrative Biology; Synthetic Biology & Biotechnology

**Mol Syst Biol. (2017) 13: 952**

## Introduction

Natural regulatory networks control the timing and conditions for gene expression. An ability to construct synthetic networks would enable the spatiotemporal control of biological processes (Basu *et al*, 2004). These could be used to react to environmental conditions (e.g., different phases of growth in a bioreactor; Anderson *et al*, 2006; Gupta *et al*, 2017) or implement a dynamic response (e.g., avoiding the accumulation of toxic intermediates; Zhang *et al*, 2012). However, there are many challenges when building synthetic regulatory networks. Obtaining a desired response requires numerous interacting genes and precise control over their expression. This results in large systems that contain many genetic parts, all of which must function correctly in concert. While fluorescent reporters have been critical for quantifying the response of such systems to date (Kelly *et al*, 2009), they are only capable of probing a single gene at a time (usually the output) and require repetition of the assay for each state or time point of interest. Mapping the fluorescence data of the output back to the specific internal failure can be difficult or impossible.

Analogies to electronic circuits are often made when describing the computational operations performed by a regulatory network. Such "genetic circuits" have been built that function as logic gates (Anderson *et al*, 2006; Moon *et al*, 2012; Qi *et al*, 2012; Siuti *et al*, 2013; Nielsen & Voigt, 2014) as well as dynamic (Elowitz & Leibler, 2000; Zhang *et al*, 2012) and analog (Daniel *et al*, 2013) circuits. Larger circuits can be built by connecting simpler gates (Moon *et al*, 2012; Kiani *et al*, 2014; Nielsen & Voigt, 2014). This process is facilitated by defining the signal between gates as the RNA polymerase (RNAP) flux (Canton *et al*, 2008). In practice, this is achieved by designing gates such that their inputs and outputs are both promoters. The response function can then be defined as how the output promoter activity changes as a function of the input promoter activity at steady-state (Weiss, 2001; Nielsen *et al*, 2016). Gate response functions can be used to computationally predict how to build a circuit (Hooshangi *et al*, 2005; Nielsen *et al*, 2016). However, the genetic context of the gates in the circuit differs from that used to measure them in isolation and this can impact their function and, in turn, lead to circuit failures (Brophy & Voigt, 2014). Therefore, it is valuable to be able to directly measure the performance of individual gates in the final context of a circuit.

Systems biology has led to new –omics tools that offer the potential to take a snapshot of the entire internal workings of a circuit with a single experiment. Transcriptomic methods, such as RNA sequencing (RNA-seq), enable the measurement of genomewide mRNA levels with nucleotide resolution (Zhong *et al*, 2009). This

1 Synthetic Biology Center, Department of Biological Engineering, Massachusetts Institute of Technology, Cambridge, MA, USA
2 Broad Institute of MIT and Harvard, Cambridge, MA, USA
  *Corresponding author. Tel: +1 617 253 8735; E-mail: cavoigt@gmail.com
  †These authors contributed equally to this work

can be used to calculate promoter and terminator strengths (Smanski *et al*, 2014; Li *et al*, 2015; Srikumar *et al*, 2015), which are closely related to the RNAP fluxes used to connect transcriptional gates. RNA-seq takes advantage of next-generation sequencing to quantify genomewide RNA levels at a moment in time. It has been used to address problems in strain and metabolic engineering (Yuan *et al*, 2011; Kim *et al*, 2012; Zhang *et al*, 2012; Woodruff *et al*, 2013), but has not been applied to the characterization of genetic circuits. This stems in part from the cost of RNA-seq and the large numbers of states and time points required to fully characterize a circuit. Another problem is that sequencing generates a deluge of data and a lack of software tools for synthetic biology, and especially genetic circuit design, hinders its processing and interpretation.

Several advances have been made that reduce the cost of RNA-seq and enable multiple circuit states to be assayed in a single sequencing run. RNAtag-seq uses nucleotide barcodes to tag total fragmented RNA before depletion of ribosomal RNA (rRNA) to allow for many samples to be efficiently pooled and sequenced together (Shishkin *et al*, 2015). The tags denote which sample a fragment originates and allows for data from each sample to be separated post-sequencing. This leads to significant reductions in the cost of reagents, accelerates preparation time, and decreases biases due to the amplification of individual sample libraries. Here, we apply this method to barcode samples associated with different states of a genetic circuit. Specifically, we characterize the eight states of a three-input one-output combinatorial logic circuit. This approach can be scaled-up: a single flow cell on an Illumina HiSeq 2500 machine generates ~4 billion paired-end reads and is thus capable of characterizing up to 1,000 samples (Haas *et al*, 2012), which could be used to simultaneously assay many different circuits and states.

Another advantage of RNA-seq is that data is captured for the entire host genome. This enables the direct observation of how differing circuit states impact host gene expression and the burden imposed by the circuit (Ceroni *et al*, 2015). It has been shown that availability and sequestering of shared cellular resources can significantly impact circuit function (Cardinale *et al*, 2013; Jayanthi *et al*, 2013; Gorochowski *et al*, 2016). Therefore, as circuits become larger, accounting for these effects will become increasingly important.

In this manuscript, we present methodologies for the application of RNA-seq to characterize genetic circuits (Fig 1). A combinatorial logic circuit is chosen as a demonstration, and data are collected for all permutations of the inputs. Cells with circuits in these different states are sequenced using RNAtag-seq and new algorithms and software are used to automate data processing. Biophysical models are developed that connect the functions of promoters, terminators, and insulators to the expected transcriptional profiles. This is used to algorithmically quantify the performance of genetic parts. Furthermore, the data are used to quantify the response functions of the three sensors and five NOT/NOR gates in the context of the circuit. This analysis reveals several mechanistic causes of circuit failures. The ability to observe the internal workings of genetic circuits will lead to a better understanding of the mechanisms that lead to part failure, how this propagates to impact system function, and ultimately will support the construction of larger genetic systems.

# Results

## Data collection

The first step in characterizing a genetic circuit is to gather data covering the range of states (Fig 1A). This differs depending on the type of circuit; for logic, this corresponds to steady-state measurements for each combination of inputs, whereas for a dynamic circuit, it would involve sampling time points. Once reaching either steady-state or a specific time point, aliquots of cells harboring the circuit are taken and flash-frozen in liquid nitrogen to minimize RNA degradation (Materials and Methods). The total RNA is then harvested, purified, and concentrated.

Samples are next converted to a pooled sequencing library using RNAtag-seq (Shishkin *et al*, 2015). First, they are separately fragmented before short DNA adaptors containing unique barcode sequences are ligated to the 3′-end of the RNAs. These barcodes uniquely "tag" every molecule such that the originating sample is known. Due to end specificity of ligation, they also capture strand-specific information. To ensure later sequencing is not affected by barcode choice, we use a set where minimal sequencing bias has been observed (Shishkin *et al*, 2015). Tagged samples are pooled to simplify the remaining preparatory steps. Unwanted rRNA is depleted, and cDNA generated by reverse transcription. Then, any remaining RNA is degraded and 3′ DNA adaptors are ligated such that a final library can be produced by amplification with indexed sequencing primers. Finally, the library is sequenced to generate strand-specific reads and the barcodes are used to associate each read to its original sample. Files containing these data are then used as inputs for circuit characterization.

## Conversion of raw RNA-seq reads to transcription profiles

We developed a suite of algorithms to process RNA-seq data and characterize part performance, sensor/gate function and host response (Fig 1A). This requires the conversion of the raw sequencing data into a transcription profile for each input state or time point (Box 1). The profiles capture the observed number of transcripts at every position along the DNA encoding the circuit. To perform this conversion, the second step in our pipeline takes as input the raw reads generated from sequencing and maps these to user-provided reference sequences containing both the host genome and synthetic circuit in a multi-FASTA format using BWA (Li & Durbin, 2009; Fig 1A). Strand-specific transcription profiles are then generated by separately extracting reads mapping to the sense and antisense strand and their start and end position using SAMtools (Li *et al*, 2009). A mathematical model is then applied to correct the transcription profiles for the localized drops in sequencing depth at the ends of transcripts, using the mapped fragment length distribution as an input variable (Box 1). This correction is required to be able to characterize parts that occur near transcript start and end sites (e.g., promoters and terminators). To provide further gene-level expression estimates for the host and circuit, a user-provided sequence annotation in GFF format containing the region of each gene is used by HTSeq (Anders *et al*, 2015) to count the reads mapping to each gene.

Because RNA-seq provides relative measurements of transcript abundance (Robinson & Oshlack, 2010; *i.e.*, fractional abundance of

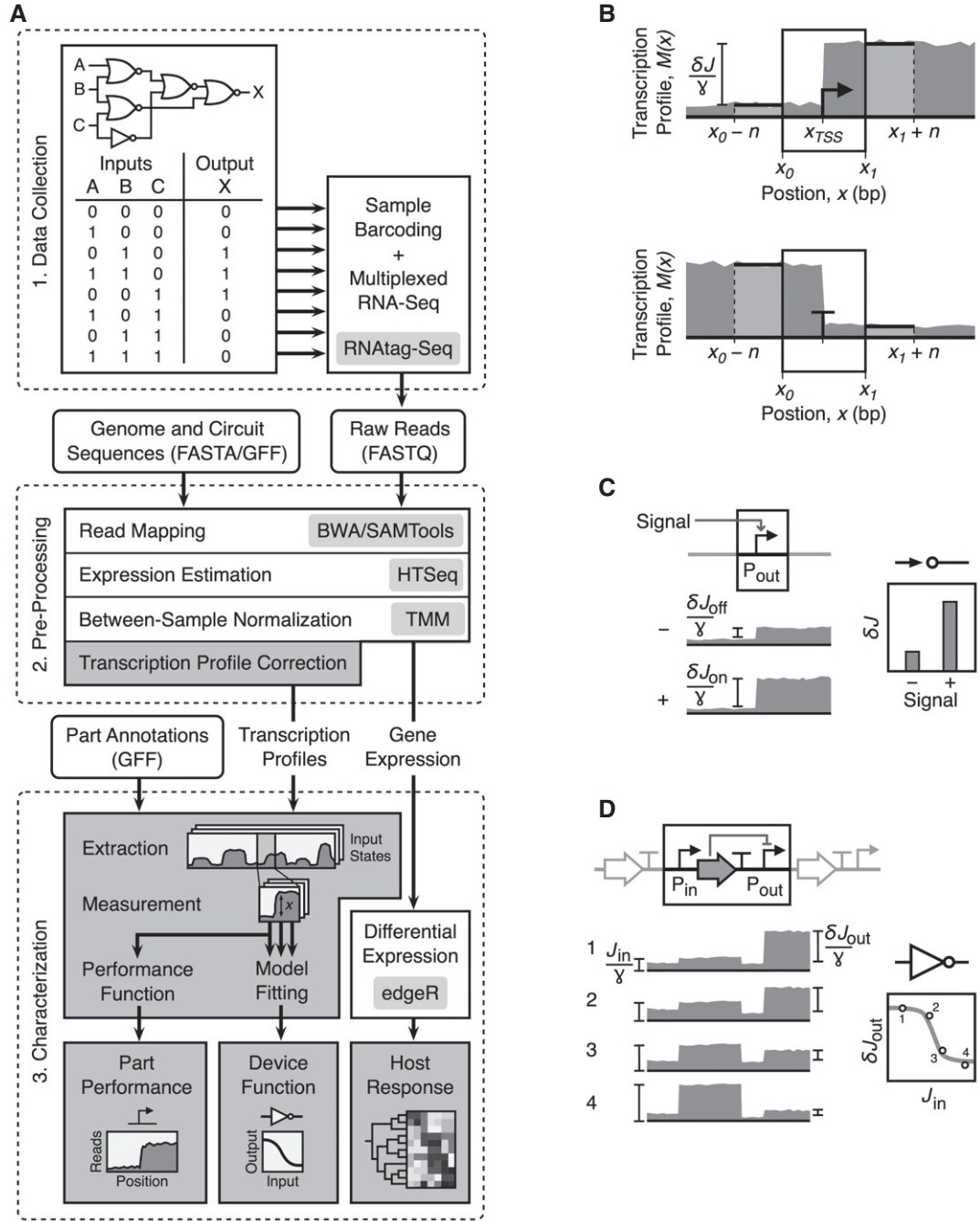

**Figure 1.  Overview of RNA-seq circuit characterization.**

A   Circuit characterization pipeline. Square boxes are the major steps in the process, and rounded boxes are input/output files. Light gray boxes denote experimental protocols or computational tools used during that process. Dark gray boxes correspond to the algorithms developed in this work and the major outputs from the pipeline. Details regarding the software to process sequencing data are provided in the Materials and Methods.

B   Quantification of the performance of promoters and terminators from transcription profiles. Transcription profiles are shown in dark gray, with the location and extent of the promoter and terminator shown by a box. Parameters correspond to equations 2 and 3.

C   Characterization of sensors. The activity of the output promoter is measured in the absence and presence of the associated signal.

D   Characterization of gates. Measurements of the total input RNAP flux and the change due to the promoter are measured for each state, and these data are used to parameterize the response function. Parameters correspond to equation 4.

the total sequenced fragments), potential differences introduced during sample preparation or sequencing require correction to allow for comparison between samples. This is performed by calculating between-sample normalization factors using a trimmed mean of M-values (TMM) approach (Robinson & Oshlack, 2010). These factors are applied to produce normalized gene expression levels in

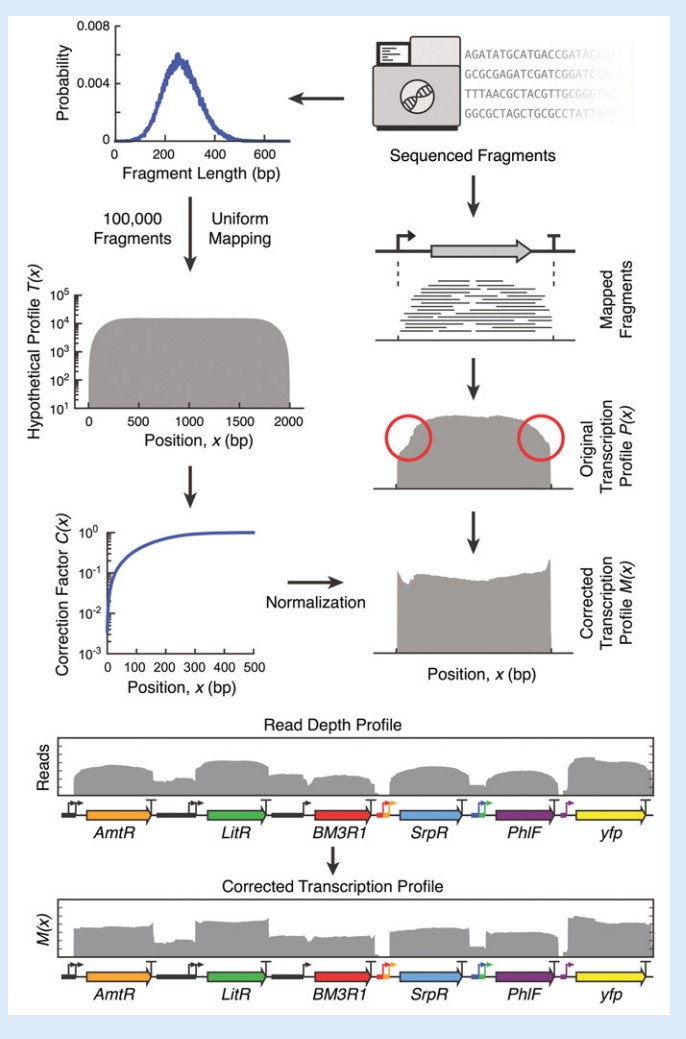

**Box 1:  Generation of transcription profiles for part characterization**

A widely used approach to generate transcription profiles from RNA-seq data is to count the number of mapped fragments that cover each nucleotide position along the DNA sequence (Nagalakshmi *et al*, 2008; Zhong *et al*, 2009). The resulting profiles often exhibit a gradual but significant increase and decrease at the start and end of transcription units, respectively. These non-uniformities (shown by red circles) are problematic for characterizing the performance of genetic parts such as promoters and terminators, where their function is defined by the regions near the transcription start and end points.

To address this problem, a probabilistic method was developed to correct for curvature at the ends of each transcription unit, which utilizes the fragment length distribution as the only input variable. We begin by generating a fragment length distribution by directly analyzing the sequenced fragments. Using a Monte Carlo approach, large numbers of fragment lengths are drawn from this distribution. Fragments of these lengths are randomly mapped to positions falling within the boundaries of a hypothetical transcription unit 2,000 nt long. By counting the number of mapped fragments covering each nucleotide $x$, a hypothetical profile $T(x)$ is produced that defines the expected curvature at each end of a transcription unit. Because the curvature is localized and fully captured by the first 500 nt of the hypothetical profile, this region is extracted and normalized by its maximum value to generate a correction factor profile $C(x)$.

Next, the RNA-seq data are used to generate the transcription profile for each transcription unit within the circuit. Fragments mapping exclusively within transcription unit boundaries are selected and a transcription profile $P(x)$ generated by counting the number of mapped fragments covering each nucleotide. Unwanted curvature is corrected for by dividing the value of $P(x)$ for the first and last 500 nt of each transcription unit by $C(x_n)$, where $x_n$ is the distance in nucleotides to the nearest end of the transcript. Specifically, the corrected transcription profile is given by

$$P_c(x) = \begin{cases} \frac{P(x)}{C(x_n)}, & 0 < x_n \leq 500, \\ P(x), & \text{otherwise} \end{cases}$$

Because the correction factor is only based on the length of the fragments and does not consider their sequence, the same correction factor profile is used for both the 5′- and 3′-end, while the middle of a transcription unit, where no curvature is present, remains unmodified. The correction factor is only applied to known transcripts internal to the genetic circuit. It is not applied to other regions where part function does not have to be calculated; for example, read-through between transcriptional units, transcripts from internal promoters, antisense transcription, and genomic transcription. Further details regarding the correction method are provided in Appendix Text S1.

fragments per kilobase of exon per million fragments mapped (FPKM) units (Trapnell *et al*, 2010) and to the transcription profiles to enable comparisons between samples (Appendix Text S1).

## Genetic part characterization from transcription profiles

Genetic parts, including promoters and terminators, impact the shape of the transcription profile by altering the flux of RNAP and mRNA transcripts produced. Techniques have been developed in bioinformatics and systems biology to naively scan transcription profiles for natural regulatory features, such as promoters (Conway *et al*, 2014). Analyzing a synthetic system has different objectives that require a different computational approach. First, the parts are modular and defined with clear start and end points. Second, the function of a part needs to be quantified (even if it is non-functional) and doing so requires a biophysical model that can

process RNA-seq data. Here, we develop models to characterize those parts that are the most critical in the design of transcriptional genetic circuits. Regions of the transcription profiles corresponding to each part are extracted, and measurements of localized changes in the profile depth are taken. These are interpreted in the context of biophysical models of each part type to infer their performance.

Promoters cause sharp increases in the transcription profile (Fig 1B). The activity of a promoter can be quantified as the increase in the flux of RNAPs that occurs between the beginning $x_0$ and end $x_1$ of a part. The RNAP flux $J(x)$ is the number of RNAPs passing nucleotide position $x$ per second. Here, we assume that all RNAPs that pass a nucleotide lead to an mRNA transcript and that all transcripts within the circuit degrade at the same rate. With these assumptions, the flux at a position $x$ is given by the steady-state number of transcripts $M(x)$ at that position (in effect, counting the

number of RNAPs passing that position that occur on the timescale of degradation).

The transcription profile provides the steady-state number of transcripts $M(x)$ at each position $x$. However, the profiles cannot be quantified in units of transcripts and are thus presented in arbitrary units (au) and the fluxes are in au/s. The change in the number of transcripts at position $x$ is given by

$$\frac{dM(x)}{dt} = J(x) - \gamma M(x), \tag{1}$$

where $\gamma = 0.0067$ s$^{-1}$ is the degradation rate of mRNA (Chen *et al*, 2015). At steady state, $dM(x)/dt = 0$ and the flux of RNAP $J(x) = \gamma M(x)$. The activity of a promoter can be quantified as the change in flux $\delta J$ that occurs over the length of the part (note that a promoter part could have multiple transcription start sites, $x_{TSS}$). To reduce the effect of fluctuations in the profile, an averaging window is applied immediately before and after the part boundaries (Fig 1B). The promoter strength is then given by

$$\delta J = \frac{\gamma}{n} \left[ \sum_{i=x_1+1}^{x_1+n} M(i) - \sum_{i=x_0-1}^{x_0-n} M(i) \right], \tag{2}$$

where $n = 10$ is the window length (Fig 1B). The background RNAP flux originating upstream of the promoter is subtracted to ensure that only flux originating from the promoter is measured.

Terminators cause drops in the transcription profile at the 3′-end of the poly-A region (Fig 1B). The terminator strength $T_S$ has been previously defined as the fold decrease in gene expression before and after the terminator (Chen *et al*, 2013). Based on the profile, it can be calculated as

$$T_S = \frac{\sum_{i=x_1+1}^{x_1+n} M(i)}{\sum_{i=x_0-1}^{x_0-n} M(i)}, \tag{3}$$

where $x_0$ and $x_1$ are the beginning and end positions of the terminator part. Following the approach for promoters described above, the activity of a terminator can also be calculated as a change in flux $\delta J$ as RNAPs either dissociate from the DNA or read-through.

### Characterization of genetic devices from transcription profiles

Sensors and gates are examples of genetic devices, where a set of parts collectively performs a function. RNA-seq is particularly suitable for characterizing transcriptional devices, where the inputs and/or outputs are defined as RNAP fluxes. For example, the input to a sensor is a stimulus (e.g., inducer or environmental signal) and the output is the control of a promoter (turning RNAP flux on or off). For gates, the inputs and outputs are both promoters and the response function captures how the output changes as a function of the input at steady-state. Unlike genetic parts, whose function can be extracted from a single profile, characterizing a sensor or circuit requires sampling the device in different states, extracting the activities of the input/output promoters, and then fitting these data to a mathematical model of device performance.

The response of a sensor is given by the activity of the output promoter in the presence and absence of signal, $\delta J_{on}$ and $\delta J_{off}$, respectively (Fig 1C). This can be calculated by performing RNA-seq

experiments under these conditions and then calculating the promoter activity according to equation 2. More states with intermediate levels of inducer are required to calculate the full dose-dependent response function. In this manuscript, the circuit is characterized in multiple states, a subset of which may have the sensor in the on or off state. In these cases, we simply average the promoter activities across those states where the sensor is on and those where the sensor is off and this is presented as the response.

A transcriptional gate has one or more input promoters and a single output promoter. The response function captures how the activity of the output promoter changes as a function of the input flux (the input promoters and upstream transcriptional read-through) at steady-state. For example, a NOT gate has one input promoter that drives the expression of a repressor that turns off an output promoter. The response function of this gate is

$$\delta J_{out} = \delta J_{out}^{min} + \left( \delta J_{out}^{max} - \delta J_{out}^{min} \right) \left( \frac{K^n}{K^n + J_{in}^n} \right) \tag{4}$$

where $J_{in}$ is total input flux, $\delta J_{out}^{min}$ and $\delta J_{out}^{max}$ are the minimal and maximal output promoter activities, $K$ is threshold, and $n$ is the cooperativity. When there is no transcriptional read-through from upstream of the input promoters, then $J_{in} = \delta J_{in}$. NOR gates have a similar structure as a NOT gate, but include multiple input promoters. For two-input NOR gates, $J_{in} = \delta J_{in,1} + \delta J_{in,2} + J_0$, where the 1 and 2 subscripts indicate the activity of the two input promoters and $J_0$ is the read-through from upstream of these promoters.

RNA-seq experiments could be designed to characterize the response function of individual gates by taking samples where the inputs are varied, calculating the promoter activities from the profiles and then fitting them to a mathematical form of a response function. Here, we wanted to be able to quantify multiple gates within the context of a circuit. For example, when characterizing combinatorial logic, the sensors are induced in all combinations (e.g., a three-input logic gate has eight combinations of inputs). Under these different conditions, the magnitude of the input promoter activity to the gate varies because of changes to the remainder of the circuit. We utilize those changes to plot data points for $J_{in}$ and $\delta J_{out}$ (Fig 1D) and this can be fitted to a response function (equation 4) to extract the parameters $\delta J_{out}^{min}$, $\delta J_{out}^{max}$, $K$, and $n$.

### Characterization of a combinatorial logic circuit

We applied our characterization method to a three-input one-output combinatorial logic circuit (Fig 2A). Three sensors respond to IPTG, aTc, and arabinose, and their activities are processed by five layered NOR/NOT gates. The complete circuit consists of 46 genetic parts (Fig 2B), including promoters, genes, terminators, and ribozyme insulators (Lou *et al*, 2012). The output of the circuit is yellow fluorescent protein (YFP), which allows for the use of flow cytometry to measure the response in single cells. Cello was used to simulate circuit performance based on the sensor and gate functions measured in isolation (Materials and Methods) (Nielsen *et al*, 2016). This circuit was selected because overall it functioned as predicted in terms of producing the correct pattern of on and off outputs, but several of the responses (+/−/+, −/+/+, +/+/+) had wide distributions indicating that some of the cells were responding improperly (Fig 2C).

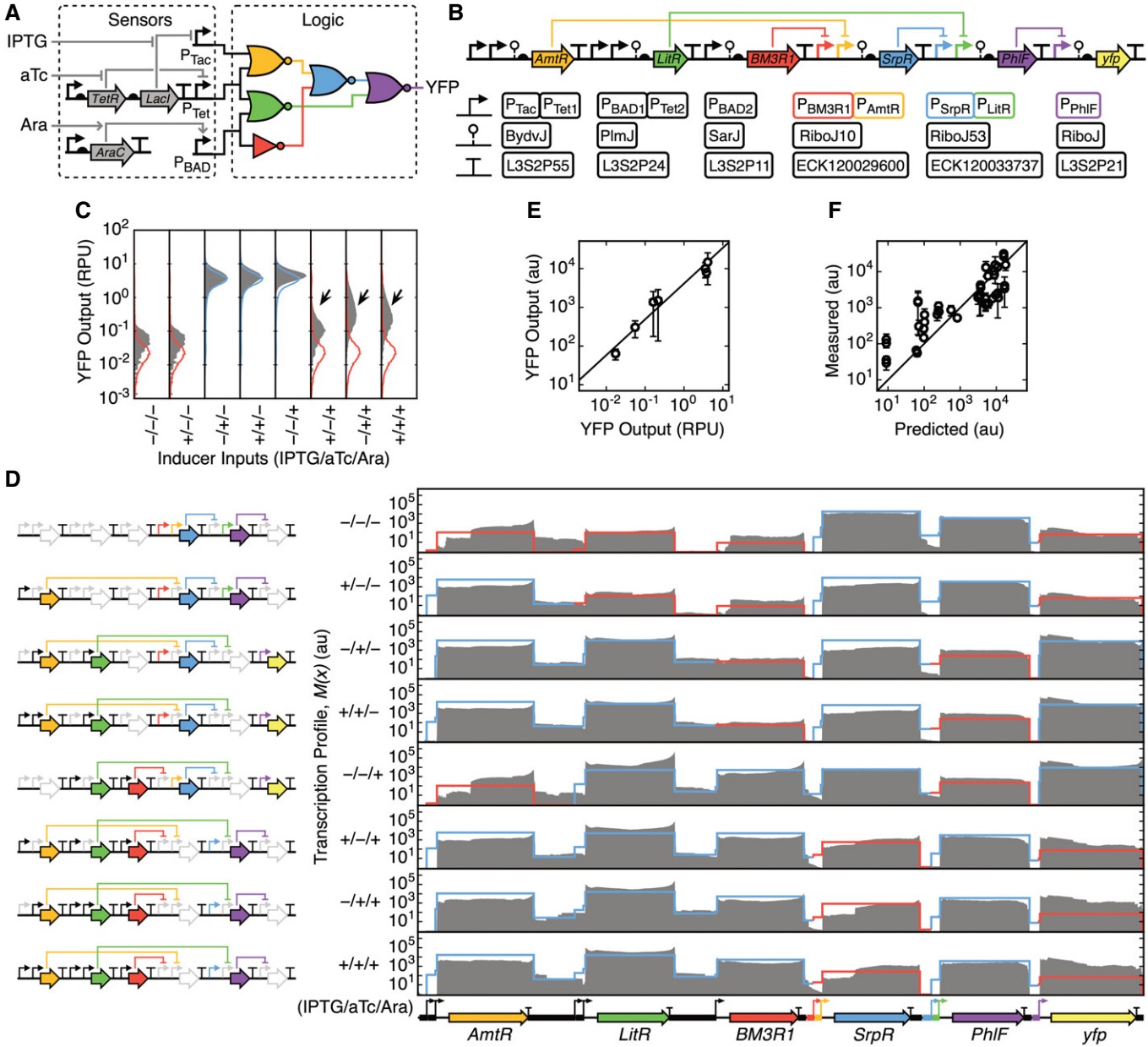

**Figure 2.  Characterization of a genetic circuit.**

A  The sensors and wiring diagram for the three-input combinatorial logic circuit are shown. Colors correspond to the repressors used for each gate.

B  Genetic implementation of the logic circuit with annotated part names. Genetic parts are shown using SBOLv notation.

C  Flow cytometry data of the YFP output for all inducer input combinations (filled gray distributions) and the predicted distributions from Cello (blue are on; red are off). Arrows highlight cells responding improperly.

D  Transcription profiles for the circuit are shown for all combinations of inducers (0.5 mM IPTG/22 nM aTc/5 mM arabinose). The transcription profiles are calculated as the average of three biological replicates measured on different days. Predicted transcription profiles are shown by red and blue lines corresponding to when the gate should be off and on, respectively, as calculated using Cello (Materials and Methods). To the left of the profiles, the genes that are expressed in each state are highlighted.

E  Determination of the conversion factor between RPU measured via cytometry and the expression of the *yfp* gene measured by RNA-seq. The black line shows the linear fit. The averages and standard deviations were calculated from three replicates measured on different days.

F  Comparison of the expression of circuit genes predicted by Cello and measured experimentally from the transcription profile (Materials and Methods). Black line shows *x* = *y*. The averages and standard deviations were calculated from three replicates measured on different days.

The states of a combinatorial logic circuit are defined as the steady-state responses to all combinations of inputs. Cells containing the circuit were grown for 5 h in media supplemented with the eight combinations of inducers, and RNA samples were collected and processed (Materials and Methods). Sequencing of these samples generated between $2.0 \times 10^5$ and $1.8 \times 10^7$ mapped

fragments (Dataset EV1). These were pre-processed to generate transcription profiles and normalized gene expression levels. The resultant transcription profiles displayed punctuated forms with expression of the transcriptional units for each gene clearly separated in most cases (Fig 2D). Two additional replicates were performed for all input states on different days and processed in the same way, and the resulting profiles are consistent (Appendix Fig S2).

Comparing simulation predictions to the measured transcription profiles is complicated by the fact that Cello reports relative promoter units (RPUs). To convert RPUs to arbitrary units that are compatible with the transcription profiles, the activity of the YFP output was measured in RPUs and compared to the average transcription profile across the *yfp* gene. These were linearly correlated (Fig 2E) with a conversion factor of 1 RPU = 2,895 au. Cello predictions for all promoter activities were converted using this factor. These were then used to trace a predicted profile along the length of the construct (Fig 2D; Materials and Methods). A correlation was found between the predicted and experimental transcription profiles for each gene (Fig 2F). Note that this is the first time we have been able to compare the levels of the repressors to that predicted; previously, the correlation could only be quantified for the output (YFP).

Next, every promoter and terminator in the circuit was characterized across each of the eight states. An idiosyncrasy in our gate designs required a modification to the approach to characterize promoters. Normally, the rise in the transcription profile would be observed just after the promoter part. However, we use ribozymes as part of our gate design to insulate against contextual effects caused by changing the upstream promoters (Lou *et al*, 2012). The ribozymes cleave the 5′-UTR sequences, releasing a small RNA that is filtered during sequencing preparation (Appendix Text S2; Appendix Fig S3). This causes the increase in the transcription profile

to occur at the cleavage site of the ribozyme (Fig 3A; Appendix Fig S4). To calculate the promoter activities when there is a ribozyme, equation 2 is changed so that the fluxes are calculated downstream of the cleavage site. A mathematical model of ribozyme efficiency was also constructed that can be used to quantify imperfect cleavage efficiencies (Appendix Text S2; Appendix Fig S4; Appendix Table S1).

The use of ribozymes also makes it impossible to resolve the individual activities of multiple promoters in series (because the cleaved 5′-RNA from transcripts from either promoter are lost during processing). Due to the use of multiple NOR gates, there are many examples of this in the circuit (e.g., $P_{Tac}-P_{Tet1}$). Therefore, we calculate the dual promoter as a single promoter part using equation 2. If sufficient data are generated across states, the contributions of the individual promoters to the total can be deduced (see next section).

The transcriptional profiles for all of the promoters in the circuit are shown in Fig 3A, and the activities are provided in Table 1. The activities of the promoters for all eight input states are shown, which includes cases where the promoters should be off and on. The values calculated from the profiles are compared to the activities measured in isolation using fluorescent proteins (Table 1). Terminator strengths were calculated from the drops in the profile at the end of transcripts (Fig 3B) and compared to strengths measured in isolation using fluorescent reporters (Table 1).

## Characterization of devices internal to the circuit

The circuit contains eight genetic devices: three sensors, four NOR gates, and a NOT gate. Each was characterized in isolation by empirically measuring the response function using a fluorescent reporter (Nielsen *et al*, 2016). This information was then used by Cello to predict how to connect them to build the larger circuit.

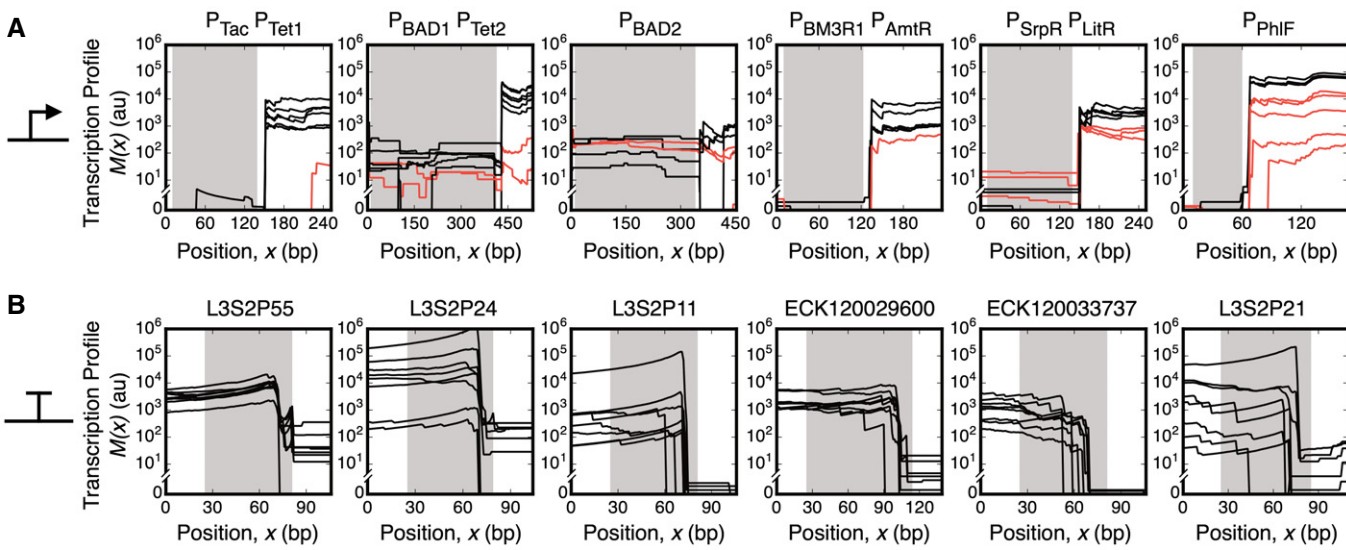

**Figure 3.  Quantifying part function.**

A, B  Alignment of transcription profiles for promoters (A) and terminators (B). Lines show the transcription profile for each of the eight combinations of inducers. Shaded regions denote the boundaries of the part. The lines are black for states where the promoter should be on and red when it should be off. The data are shown for the profiles derived from a single experiment (part parameterization from three replicates is provided in Table 1).

Here, we calculated the performance of the devices in the context of the circuit using RNA-seq and compare the values to their measurement in isolation using a fluorescent protein and cytometry.

The sensor response function is specified by the activity of the output promoter in response to the presence and absence of an inducer. When a sensor occurs alone (e.g., $P_{BAD2}$), we used the measured promoter strengths (Table 1) to calculate $\delta J_{off}$ and $\delta J_{on}$ directly. When the output promoters appear in pairs (e.g., $P_{Tac}-P_{Tet1}$ and $P_{BAD1}-P_{Tet2}$), the promoters cannot be individually resolved directly from the RNA-seq data. To separate the activities of the two promoters, the individual outputs of the first and second promoters are defined as $\delta J_{1,on}$ and $\delta J_{2,on}$, or $\delta J_{1,off}$ and $\delta J_{2,off}$, where on and off refers to the presence or absence of the inducer. Then, for each state, the measured combined activity of the promoter pair $\delta J_{1+2}$ is equated to the expected activities for different combinations of inducers. For example, when the inducers that activate both sensors are present, $\delta J_{1+2} = \delta J_{1,on} + \delta J_{2,on}$. This yields a set of algebraic equations to which $\delta J_{1,on}$, $\delta J_{1,off}$, $\delta J_{2,on}$, and $\delta J_{2,off}$ can be fitted (Materials and Methods).

Sensor responses in the context of the circuit can then be compared to the values measured for the sensor in isolation (Fig 4A; Table 2). Strong context effects were observed for $P_{BAD}$ (Table 2). In one location in the circuit, it performs as expected ($P_{BAD1}$). However, $P_{BAD2}$ produces a lower than expected on state, which manifests as a nearly flat transcription profile (Fig 3A). This coincided with read-through from the highly expressed *LitR* gene upstream (Fig 2D).

**Table 1.  Promoter and terminator part characterization.**

| | Genetic context | | | |
| | Isolation (Cytometry)[b] | | Circuit (RNA-seq)[c] | |
| Promoters[a] | 1st | 2nd | 1st | 2nd |
|---|---|---|---|---|
| $P_{Tac}$-$P_{Tet1}$ | 54 | 85 | 8 ± 2 | 17 ± 5 |
| $P_{BAD1}$-$P_{Tet2}$ | 49 | 85 | 242 ± 118 | 283 ± 237 |
| $P_{BAD2}$ | 49 | | 26 ± 29 | |
| $P_{BM3R1}$-$P_{AmtR}$ | 10 | 75 | 9 ± 4 | 55 ± 32 |
| $P_{SrpR}$-$P_{LitR}$ | 41 | 85 | 37 ± 20 | 14 ± 4 |
| $P_{PhlF}$ | 80 | | 323 ± 25 | |
| **Terminators** | | | | |
| L3S2P55 | 418 | | 24 ± 6 | |
| L3S2P24 | 212 | | 295 ± 176 | |
| L3S2P11 | 384 | | 110 ± 89 | |
| ECK120029600 | 374 | | 380 ± 98 | |
| ECK120033737 | 391 | | 870 ± 344 | |
| L3S2P21 | 505 | | 187 ± 127 | |

[a]The strength of the left promoter is 1st; right is 2nd.
[b]Previously reported promoter strengths (in au/s) based on a fluorescent reporter (Nielsen *et al*, 2016), converted to au/s as described in text. Previously reported termination strengths (Chen *et al*, 2013).
[c]Average and standard deviations are calculated from three replicates performed on different days. For promoters, all states where the promoter is predicted to be on are included and the units are au/s. For double promoters, separate strengths for each promoter are calculated as described in the text. Median terminator strengths are calculated for states where the upstream gene is on. For terminators L3S2P24 and L3S2P11 in one replicate, the data for input state −/−/+ were excluded due to a mapping bias (Appendix Fig S1).

The internal gates were characterized by calculating the full response functions from the transcription profiles for all eight states. NOR gates are characterized as NOT gates by having the multiple upstream input promoters serve as the combined input to the gate. The promoter activities were calculated from the profiles and fit to equation 4. The total flux into the gate serves as the input $J_{in}$ (either one or two promoters and upstream read-through), and the output is the activity of the output promoter $\delta J_{out}$. It is simple to calculate when this appears as a single promoter. If it is part of two promoters in series, the individual promoter activities are calculated as described for sensors, above (Materials and Methods).

The calculated $J_{in}$ and $\delta J_{out}$ values for the eight states are fit to equation 4 to obtain the response functions for each gate in the circuit (Fig 4B). The response functions (solid lines, Fig 4B) and parameter values (Table 2) are compared to those obtained from the gate measurements performed in isolation using fluorescent reporters (dashed lines, Fig 4B). The performance of the five gates is similar with several notable exceptions. The LitR gate displayed lower output flux for both on and off states, while the PhlF gate saw elevated output flux and a > 2-fold shift in the input flux required to switch the gate into an off output state.

## Part substitution to correct antisense transcription

All genes in the circuit are organized on the sense strand (Fig 2B), but the RNA-seq data also report transcription in the antisense direction (Fig 5A; Appendix Fig S5). There are several mechanisms by which antisense transcription can interfere with the function of the circuit (Shearwin *et al*, 2005; Brantl, 2007; Brophy & Voigt, 2015). Most reads corresponding to antisense transcription cluster within the *AmtR*, *LitR*, and *BM3R1* genes, implying the existence of reverse promoters internal to these genes. The $P_{BAD}$ promoters we used have known antisense transcription start sites (Schleif, 2003), from which antisense transcription could be observed (Fig 5A).

A part substitution was made to correct for the observed anti-sense transcription. In the original circuit design, the terminators only stop RNAP coming from the sense direction. One mechanism to stop antisense termination is to use a bidirectional terminator (Chen *et al*, 2013) or to follow a terminator with a terminator oriented in the opposite direction. To this end, we replaced the terminator after the *LitR* gene with two terminators, each of which blocks transcription in opposite directions. This completely blocked antisense transcription (Fig 5B), demonstrating the use of RNA-seq to rationally correct an observed error in a circuit.

## Impact of circuit state on host gene expression

Different combinations of inputs will cause different genes in the circuit to be expressed (Fig 2D), and this can change the burden on the host cell (Ceroni *et al*, 2015). For the three-input logic circuit, an increase in cell doubling times is observed for input states where four genes (including repressors and *yfp*) were expressed (gray bars, Fig 6A). For the input states expressing only two and three repressors (−/−/− and +/−/−), the growth rates were similar. Input states with the slowest growth rates (+/−/+,−/+/+, +/+/+) also corresponded to those with the broadest flow cytometry distributions, deviating from the Cello predictions (Fig 2C).

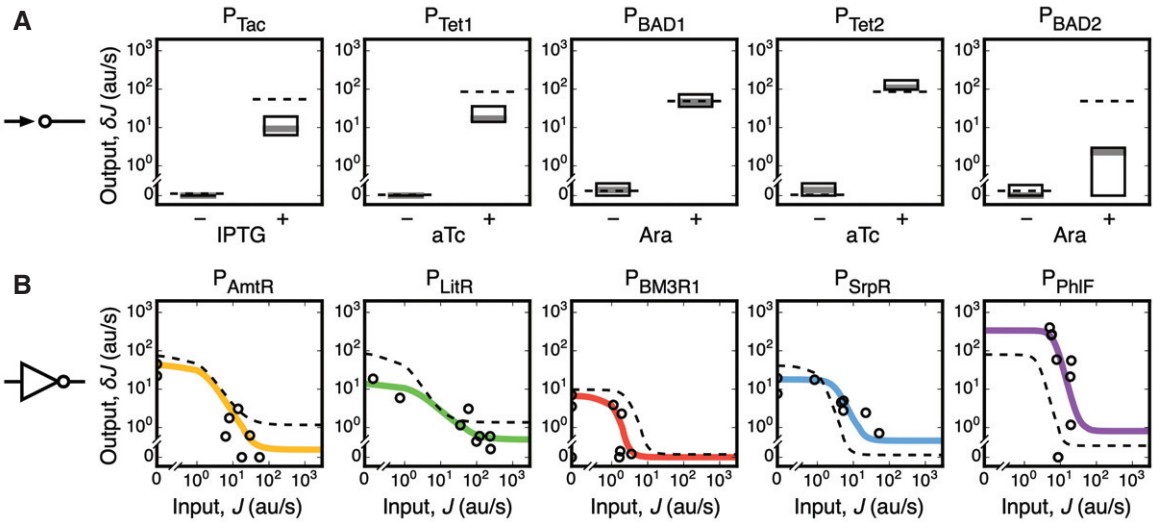

**Figure 4. Extraction of sensor and gate response functions from the transcription profiles.**

A  The responses of the output promoters of the sensors are shown in the presence and absence of each inducer. The dashed lines show the sensor outputs measured in isolation (Nielsen *et al*, 2016). The boxes show the median (gray line) and range of promoter activities measured for the four states where it is off ($\delta J_{off}$) and four where it is on ($\delta J_{on}$).

B  Solid colored lines show the response functions of the gates obtained by fitting the promoter activities to the RNA-seq data (circles denote the measured values for the eight input states). The dashed lines show the output of the gate measured in isolation (Nielsen *et al*, 2016). The data are shown for the profiles derived from a single experiment (device parameterization from three replicates is provided in Table 2).

**Table 2. Sensor and gate response function parameters.**

| Sensor | Genetic context | | | | | | | |
| | **Isolation (Cytometry)**[a] | | | | **Circuit (RNA-seq)**[b] | | | |
| | $\delta J_{off}$ | $\delta J_{on}$ | | | $\delta J_{off}$ | $\delta J_{on}$ | | |
| $P_{Tac}$ | 0.1 | 54 | | | $0.0 \pm 0.0$ | $9 \pm 0.9$ | | |
| $P_{Tet1}$ | 0.0 | 85 | | | $0.0 \pm 0.0$ | $16.0 \pm 4.8$ | | |
| $P_{Tet2}$ | 0.0 | 85 | | | $0.8 \pm 0.6$ | $274 \pm 165$ | | |
| $P_{BAD1}$ | 0.2 | 49 | | | $0.8 \pm 0.6$ | $165 \pm 114$ | | |
| $P_{BAD2}$ | 0.2 | 49 | | | $0.0 \pm 0.0$ | $13.0 \pm 9.2$ | | |
| **Gate** | $\delta J_{out}^{min}$ | $\delta J_{out}^{max}$ | $K$ | $n$ | $\delta J_{out}^{min}$ | $\delta J_{out}^{max}$ | $K$ | $n$ |
| $P_{AmtR}$ | 1.2 | 75 | 1.4 | 1.6 | $0.3 \pm 0.1$ | $80 \pm 48$ | $1.0 \pm 0.7$ | $1.7 \pm 0.2$ |
| $P_{LitR}$ | 1.4 | 85 | 1.0 | 1.7 | $1.9 \pm 1.2$ | $53 \pm 36$ | $2.7 \pm 0.5$ | $1.4 \pm 0.2$ |
| $P_{BM3R1}$ | 0.1 | 10 | 2.9 | 2.9 | $0.0 \pm 0.0$ | $12 \pm 8$ | $1.6 \pm 0.6$ | $2.9 \pm 0.2$ |
| $P_{SrpR}$ | 0.1 | 41 | 1.2 | 2.8 | $0.3 \pm 0.2$ | $16 \pm 2$ | $3.0 \pm 1.6$ | $2.5 \pm 0.2$ |
| $P_{PhlF}$ | 0.4 | 80 | 2.5 | 3.9 | $0.7 \pm 0.1$ | $337 \pm 2$ | $8.9 \pm 1.9$ | $3.6 \pm 0.2$ |

[a]Previously reported values (in au/s) based on a fluorescent reporter (Nielsen *et al*, 2016), converted to au/s as described in the text. The units of $\delta J_{off}$, $\delta J_{on}$, $\delta J_{out}^{min}$, $\delta J_{out}^{max}$, and $K$ are au/s.
[b]The units of $\delta J_{off}$, $\delta J_{on}$, $\delta J_{out}^{min}$, $\delta J_{out}^{max}$, and $K$ are au/s. Average and standard deviations are calculated from three replicates measured on different days.

The impact on the host of carrying a circuit can be observed as changes in the expression of native genes. To generate a baseline for comparison, we preformed duplicate RNA-seq experiments using cells harboring the circuit backbone (pAN1201), but without the remainder of the circuit. These data were used for differential gene analysis to search for potential differences in host gene expression. Expression of endogenous genes in cells containing the circuit for input states expressing only two or three genes (−/−/− and +/−/−) was highly correlated with the baseline ($R^2 = 0.83$; Fig 6B). For input states expressing four genes, we found a lower correlation to the baseline ($R^2 = 0.55$), and 125 significantly differentially expressed genes ($P < 0.01$; Dataset EV2). Most of these were downregulated ($N = 106$) with enrichment for functions broadly related to energy generation, anaerobic respiration, and

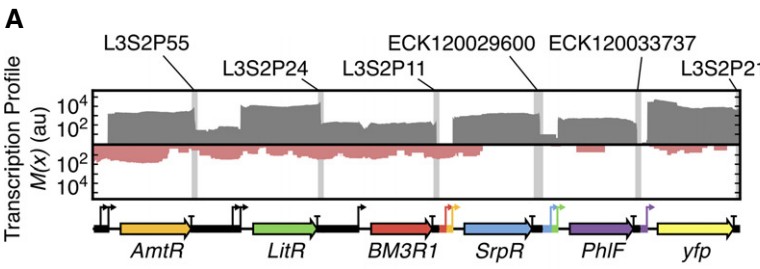

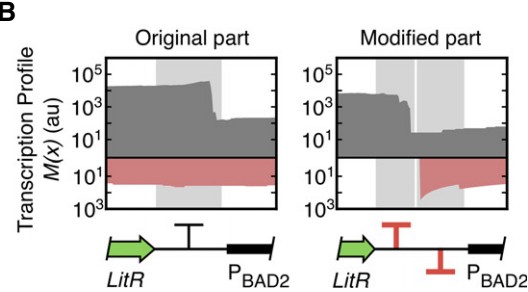

**Figure 5.  Antisense transcription, measurement, and correction.**

A  Transcription profiles for both sense (gray) and antisense (red) strands for circuit grown in culture tube conditions for the −/+/− combination of inducers (22 nM aTc). Terminators are shown by light gray-labeled regions. The antisense profiles for all combinations of inducers are shown in Appendix Fig S5. The profiles correspond to a single experiment that is representative of three replicates.

B  The change in the transcription profile that occurs due the addition of a reverse terminator is shown. Shaded regions denote terminator part boundaries. The original terminator is L3S2P24, and this is replaced by the terminators ECK120033736 (forward) and ECK120010818 (reverse). The profile normalization (Box 1) has been applied to the 3′-end of the antisense profile in the modified part.

fermentation (Appendix Table S2). Of those genes upregulated, there was significant enrichment for DNA replication and repair, iron assimilation and homeostasis, and functions linked to colanic acid biosynthesis, the production of which occurs in response to low temperature, osmotic shock, and desiccation (Navasa *et al*, 2009).

### Environmental robustness

Growth conditions impact host physiology (Kram & Finkel, 2014), and this can influence the performance of genetic parts and devices (Moser *et al*, 2012; Gorochowski *et al*, 2014). The experiments described so far were performed in 14-ml culture tubes. However, we observed serendipitously that the circuit failed when cells were grown in the same media in Erlenmeyer flasks (Materials and Methods). Under these conditions, three of the input combinations (+/−/+,−/+/+ and +/+/+) resulted in much broader flow cytometry distributions of the YFP output with an incorrect on output state for most cells (Fig 6C). This coincided with increases in the doubling times when arabinose was present (Fig 6A).

Transcription profiles were compared between cells grown in Erlenmeyer flasks and culture tubes (Fig 6D). For the working states, there are only minor differences, but for the three broken states, there are major differences. Using these data, we re-quantified part and device performance (Appendix Figs S6 and S7; Appendix Tables S3 and S4). Few changes were observed for the gates and the IPTG ($P_{Tac}$) and aTc ($P_{Tet}$) sensors, with similar response functions across conditions (Appendix Fig S7). However, large changes in performance were found for both the arabinose sensors that displayed ~2-fold increases in their induced activities. This change propagates through the circuit and impacts the levels of the repressors, culminating in a > 2-fold decrease in PhlF, which increases YFP expression (Fig 6D), thus causing the average of the cytometry populations to appear to be on for these states (Fig 6C). The level of antisense transcription is also higher from the reverse promoter within $P_{BAD}$ under these conditions (Appendix Fig S5).

We analyzed the host transcriptome under both conditions to ascertain whether a shift in cellular physiology might be the cause for the failure of the arabinose sensors ($P_{BAD1}$ and $P_{BAD2}$).

Differential expression analysis of the broken input states highlighted significant changes for 179 genes ($P < 0.01$), with enrichment of transport-related functions for arabinose, xylose, and maltose ($P < 5.69 \times 10^{-3}$; Fig 6E; Dataset EV2). Upregulation of these genes coincided with the presence of arabinose, suggesting its role in their regulation (Fig 6E). Notably, host genes involved in arabinose transport (*araEFGH*) saw significant eight- to 56-fold upregulation in culture tubes for the three broken input states ($P < 5.75 \times 10^{-4}$; Dataset EV2). Such a difference would facilitate greater intracellular accumulation of arabinose due to an increased uptake and is consistent with the large measured increases in activity of both $P_{BAD}$ promoters when induced (Khlebnikov *et al*, 2001).

## Discussion

DNA synthesis and assembly methods enable the construction of large genetic circuits that can implement complex functions through the layering of simple transcriptional gates. Gates can be built based on many classes of biochemistry (e.g., DNA-binding proteins (Moon *et al*, 2012; Stanton *et al*, 2014), recombinases (Bonnet *et al*, 2013; Siuti *et al*, 2013; Fernandez-Rodriguez *et al*, 2015), and CRISPRi (Gilbert *et al*, 2013; Larson *et al*, 2013; Qi *et al*, 2013; Kiani *et al*, 2014; Nielsen & Voigt, 2014; Gander *et al*, 2017)). Orthogonal libraries of these parts enable many to be reliably used in a single cell without fear of interference. These advances have led to circuits that can consist of 10+ regulators and > 40 genetic parts and the size is growing. However, the ability to debug systems of this size has lagged, particularly when the function is defined by many states. Here, we have developed methodologies to characterize the inner workings of a circuit using RNA-seq data.

Circuit characterization has been limited by the use of fluorescent reporters to provide a single datum for the output of the circuit as a whole. One way to measure the response of internal parts and gates is to separate them onto characterization plasmids that can be assayed in isolation of the remainder of the circuit (Kelly *et al*, 2009; Stanton *et al*, 2014; Nielsen *et al*, 2016). In contrast, RNA-seq enables the function of multiple parts to be simultaneously measured *in situ* within a circuit. Using this approach, we have

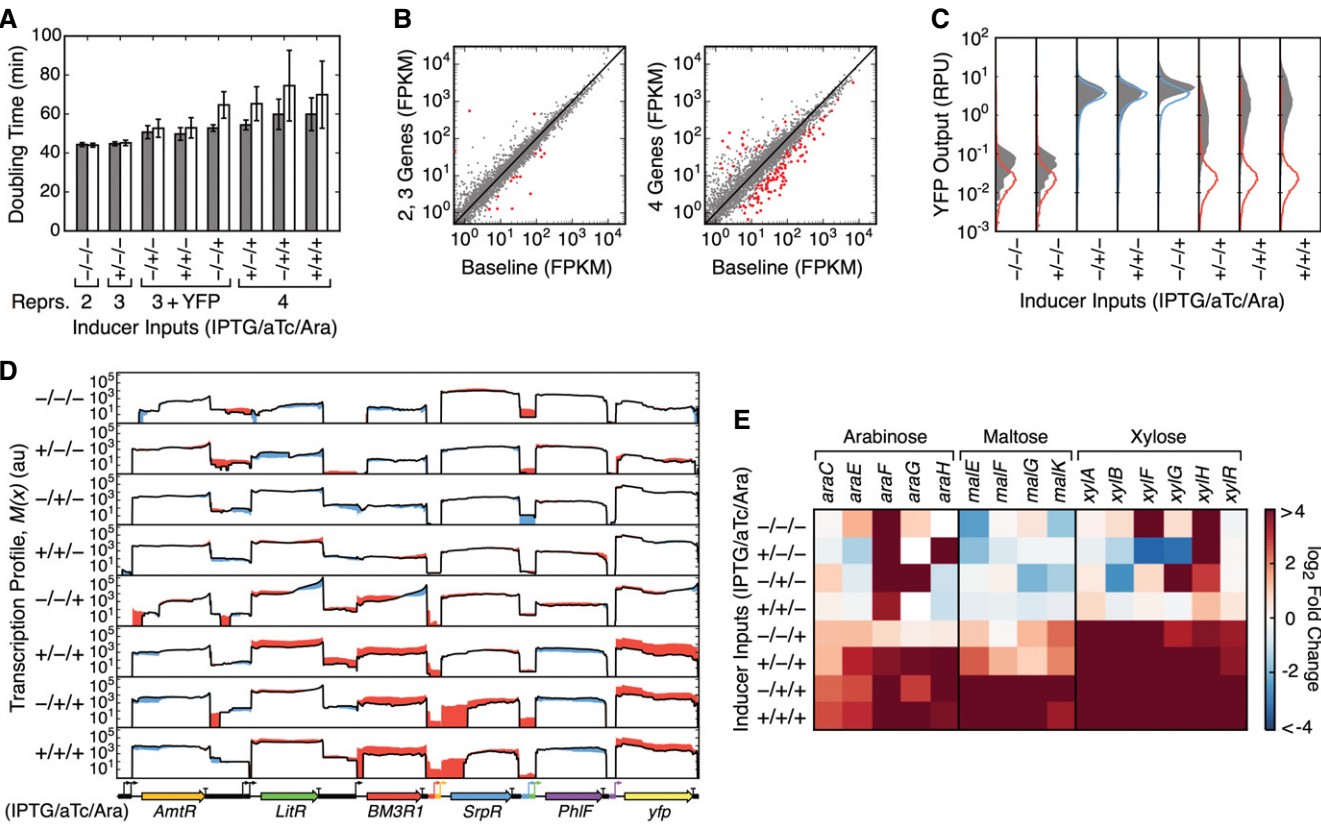

**Figure 6.    Changes to the growth conditions break the circuit by changing host gene expression.**

A    Doubling times for cells carrying the circuit grown in culture tubes (gray) and Erlenmeyer flasks (white). Error bars are calculated as the standard deviation from three biological replicates performed on different days.

B    Comparison of host gene expression under culture tube conditions for sets of input states with differing numbers of expressed circuit genes (including *yfp*): two = −/−/−, three = +/−/−, and four = −/+/−, +/+/−, −/−/+, +/−/+, −/+/+ and +/+/+ (0.5 mM IPTG/22 nM aTc/5 mM arabinose). The baseline is calculated from RNA-seq data collected from cells harboring an empty circuit plasmid backbone (pAN1201). Points show the mean expression level of each gene. Red points denote genes with statistically significantly differential expression in comparison with the baseline (*P* < 0.01; Dataset EV2).

C    Flow cytometry data of the YFP output for the circuit grown in Erlenmeyer flasks (filled gray distributions) and the predicted output distributions from Cello (blue is on and red is off).

D    Comparison of transcription profiles for the circuit when cells are grown in culture tubes (black line; data for biological replicate 1) and Erlenmeyer flasks. If the profile obtained from Erlenmeyer flasks is higher the difference is shown in red and if it is lower the difference is shown in blue.

E    Fold-change analysis of differentially expressed sugar transport-related genes (*P* < 0.003) compared between Erlenmeyer flask and culture tube growth conditions (Dataset EV2).

revealed several failure modes and showed that some parts functioned unreliably. For example, the P$_{BAD}$ promoter was particularly context dependent and sensitive to shifts in culturing conditions.

One limitation of RNA-seq is that it only measures mRNA levels. This is useful when characterizing transcriptional circuits. However, some parts such as ribosome binding sites (RBSs) and bi-cistronic insulators (Mutalik *et al*, 2013) operate at the level of translation. It has been shown that ribosome profiling (Ingolia *et al*, 2009; Ingolia, 2014) can accurately estimate protein synthesis rates (Li *et al*, 2014) and therefore could be used to characterize translational parts such as ribosome binding site strengths, if they are the rate-limiting step during protein synthesis (Li *et al*, 2014). This offers a powerful complementary technique that could, along with RNA-seq, fully characterize all of the parts in a circuit. Another limitation is that the RNA-seq method that we use is based on population measurements. Expression can differ between cells, and it is useful to have

the population data provided by cytometry or microscopy (Rosenfeld *et al*, 2005). RNA-seq can be used to measure expression in single cells, but current techniques could not be used to extract part performance data (Tang *et al*, 2009; Shapiro *et al*, 2013; Grün *et al*, 2014; Lasken & McLean, 2014).

Much of the mystery of genetic engineering comes from a lack of being able to see what you are doing; that is, how design choices impact the system and cell. Problems that can be seen are simple to correct. For example, once unwanted antisense transcription was detected, it could be blocked easily by adding a reverse terminator. If a problem cannot be seen, then this necessitates the creation of a large library of designs where many potential fixes are tried randomly until one blindly solves the problem (Smanski *et al*, 2014). New techniques to quantify mRNA, protein, and metabolites —driven by plummeting DNA sequencing costs—are giving a fuller picture of the cell. Furthermore, the routine sequencing of

laboratory strains can show how genomic mutations that emerge can impact the system (Lamb *et al*, 2006; Fernandez-Rodriguez *et al*, 2015; Song *et al*, 2017). Collectively, these are quickly clearing the fog and hand-waving underlying cellular design. However, this is also leading to a deluge of data that are increasingly perplexing to the designer. Fully utilizing these datasets will require new software that simplifies the process of collection, processing, merging data from diverse techniques, and learning. This will help reveal the design principles that allow for robust part function and support the effective development of increasingly complex genetic systems.

# Materials and Methods

### Strain, media, and inducers

The *Escherichia coli* DH10B derivative NEB 10-beta Δ(ara-leu) 7697 araD139 fhuA ΔlacX74 galK16 galE15 e14-φ80dlacZΔM15 recA1 relA1 endA1 nupG rpsL (StrR) rph spoT1 Δ(mrr-hsdRMS-mcrBC) was used for cloning and measurements (New England Biolabs, MA, C3019). Cells were grown in LB Miller broth (Difco, MI, 90003-350) for harvesting plasmids. Cells were grown in MOPS EZ Rich (Teknova, M2105) defined medium with 0.2% glycerol for circuit performance measurements. To select for the presence of plasmids, 50 μg/ml kanamycin (Gold Biotechnology, MO, K-120-5) and 50 μg/ml spectinomycin (Gold Biotechnology, MO, S-140-5) antibiotics were used.

### Circuit induction

Chemicals used to induce input promoters were isopropyl β-D-1-thiogalactopyranoside (IPTG; Sigma-Aldrich, MO, I6758), anhydrotetracycline hydrochloride (aTc; Sigma-Aldrich, MO, 37919), and L-arabinose (Sigma-Aldrich, MO, A3256). Individual colonies were inoculated into MOPS EZ Rich Defined Medium (Teknova, CA, M2105) with 0.2% glycerol carbon source and 50 μg/ml kanamycin (Gold Biotechnology, MO, K-120-5) and grown overnight for 16 h at 37°C and 1,000 rpm in V-bottom 96-well plates (Nunc, Roskilde, Denmark, 249952) in an ELMI Digital Thermos Microplates shaker incubator (Elmi Ltd, Riga, Latvia). The following day, cultures were diluted 178-fold (two serial dilutions of 15 μl into 185 μl) into EZ Rich glycerol with kanamycin and grown under the same ELMI shaker incubator conditions for 3 h. For culture tube assays (Falcon 14 ml round-bottom polypropylene tubes; Corning, MA, 352059), cells were diluted 658-fold (4.56 μl into 3 ml) into EZ Rich glycerol with kanamycin and inducers. For Erlenmeyer flask assays (Pyrex 250 ml; Cole-Palmer, IL, 4980-250), cells were diluted 658-fold (76 μl into 50 ml) into EZ Rich glycerol with kanamycin and inducers. Eight inducer combinations were used that cover the presence or absence of 0.5 mM IPTG, 10 ng/ml aTc, and 5 mM L-arabinose. Culture tubes and Erlenmeyer flasks were then grown in an Innova 44 shaker (Eppendorf, CT) at 37°C and 250 rpm for 5 h. Finally, 40 μl of cell culture was placed into 160 μl of phosphate-buffered saline (PBS) containing 2 mg/ml kanamycin to arrest translation and cell growth. These growth-arrested cells were incubated in the PBS with kanamycin for one hour before fluorescence was measured using flow cytometry.

### Flow cytometry analysis

Fluorescence of individual cells was measured using an LSRII Fortessa flow cytometer (BD Biosciences, San Jose, CA) controlled by the BD FACSDiva software. More than 20,000 gated events were collected for each state, and analysis of flow cytometry data was performed using FlowJo (TreeStar, Inc., Ashland, OR).

### RNA-seq library preparation and sequencing

Total RNA was harvested from *E. coli* DH10B cells harboring the genetic circuit plasmid and cultured under the circuit induction assay condition described above. Cultures were spun down at 4°C, 15,000 *g* for 3 min. Supernatants were discarded after centrifugation, and cell pellets were flash-frozen in liquid nitrogen for storage at −80°C. Cells were lysed with 1 mg of lysozyme (Sigma-Aldrich, MO, L6871) in 10 mM Tris–HCl (pH 8.0) (USB 75825) supplemented with 0.1 mM EDTA (USB 15694). RNA was extracted with PureLink RNA Mini Kit (Life Technologies, CA, 12183020) and further purified and concentrated with RNA Clean & Concentrator-5 (Zymo Research, R1015) to ensure sufficient RNA concentrations (> 280 ng per sample). The purified RNA samples were analyzed using a Bioanalyzer (Agilent, CA), and Ribo-Zero rRNA Removal Kit for bacteria (Illumina, CA, MRZMB126) was used to deplete rRNA from the samples. We also checked the quality of the RNA extracted by calculating the RNA integrity number (RIN), which ranges from a value of 10 if all RNA is intact to 1 if the RNA is totally degraded (Schroeder *et al*, 2006). We only consider highly intact samples with a RIN > 8.5 (Imbeaud *et al*, 2005). Strand-specific RNAtag-seq (Shishkin *et al*, 2015) libraries were created by the Broad Technology Labs specialized service facility (SSF). A total of 16 samples (one for each of the eight inducer combinations under the two different culturing conditions) were pooled, split, and run on two separate lanes of an Illumina HiSeq 2500 as technical replicates. Both lanes were checked for quality and re-pooled before reads were de-multiplexed into the original samples. Barcode sequences were trimmed from reads before further analysis was performed.

### Processing of sequencing data

Raw reads were mapped to the host genome (NCBI RefSeq: NC_01473.1) and circuit reference sequences using BWA (Li & Durbin, 2009) version 0.7.4 with default settings. The "view" command of the SAMtools (Li *et al*, 2009) suite was then used with default settings to convert the generated SAM files into a BAM format for downstream analyses. Read counts for each host and circuit gene were carried out using the "htseq-count" command of the HTSeq toolkit (Anders *et al*, 2015) with user-defined GFF annotations of the reference sequences and the options "-s reverse -a 10 -m union". Transcription profiles were generated by first splitting each BAM file into two separate BAM files that contained reads from either the sense or antisense strands. This was achieved by filtering the complete BAM file using the "view" command of SAMtools and the filter codes 83 and 163 for sense reads, and 99 and 147 for antisense reads. Normalized FPKM values were generated from the raw gene counts by custom scripts that calculated and applied a trimmed mean of M-values (TMM) factor using edgeR (Robinson *et al*, 2010) version 3.8.6. The BAM files were also separately processed by

custom Python scripts to extract the position of the mapped reads and generate the transcription profiles. Gene expression in arbitrary units (au) that are compatible with the transcription profiles was calculated as the average of the transcription profile height along the length of a gene. Differential gene expression was performed using edgeR to calculate adjusted *P*-values using the built-in false discovery rate (FDR) correction. Characterization of promoters, ribozymes, and terminators was performed using custom Python scripts that took a GFF reference of the circuit defining all part locations, their types, and any further information (e.g., predicted cut-site for ribozymes). To ensure termination strength and ribozyme cleavage were not underreported, measurements were filtered if there was a low level of RNAP flux (< 1 au/s) entering the terminator or leaving the ribozyme. Characterization of genetic parts was performed using a window size of $n = 10$ bp in equations 2 and 3, and S1. All scripts were executed using either Python version 2.7.9 or R version 3.2.1.

### Genetic circuit design and simulations

Cello (Nielsen *et al*, 2016) version 1.0 (http://www.cellocad.org) was used for all circuit simulations. For inputs, "low RPU" values of $P_{Tac} = 0.0034$, $P_{Tet} = 0.0013$, and $P_{BAD} = 0.0082$, and "high RPU" values of $P_{Tac} = 2.8$, $P_{Tet} = 4.4$, and $P_{BAD} = 2.5$ were used and the Eco1C1G1T1.UCF.json UCF file describing the gate response functions. Genetic circuit visualizations in SBOLv format (Myers *et al*, 2017) were produced using DNAplotlib (Der *et al*, 2017) version 1.0.

### Numerical fitting

To fit the response function of a gate when the output promoter was in isolation, we defined a least squares error function $E_1 = \Sigma_{i \in S}[\log(\delta J_i) - \log(\delta J_{out,i})]^2$. Here, *S* is the set of states (e.g., combinations of inducer), $\delta J_i$ is the measured activity of the output promoter for state *i*, and $\delta J_{out,i}$ is the expected output promoter activity calculated according to the gate's response function. For NOT gates, the response function is given by equation 4 with parameters $\delta J_{out}^{min}$, $\delta J_{out}^{max}$, *K*, and *n*. For each state, to calculate the expected output promoter activity of the gate, the total input RNAP flux $J_{in}$ was extracted from the transcription profile and used as input to equation 4 (Fig 1D). The "minimize" function of SciPy version 0.15.0 and the built-in sequential least squares programming algorithm were then used to fit the parameters such that the error function was minimized.

### Fitting of promoters in series and estimating their individual activities

When the output promoters of two sensors or gates were found in series, we characterized their individual behaviors by employing an adapted error function $E_2 = \Sigma_{i \in S}[\log(\delta J_{1+2,i}) - \log(\delta J_{1,i} + \delta J_{2,i})]^2$. Here, $\delta J_{1+2,i}$ is the measured combined activity of both output promoters for state *i*, and $\delta J_{1,i}$ and $\delta J_{2,i}$ are the expected individual output promoter activities calculated using the response functions of the first and second sensor/gate, respectively. For sensors, the response function was equated to either $\delta J_{on}$ or $\delta J_{off}$ depending on the inducers that were present for that state. For example, given all

combinations of inducers for two sensors in series: −/−, −/+, +/−, and +/+, then the expected combined output activity $\delta J_1 + \delta J_2$ would be given by: $\delta J_{1,off} + \delta J_{2,off}$, $\delta J_{1,off} + \delta J_{2,on}$, $\delta J_{1,on} + \delta J_{2,off}$, and $\delta J_{1,on} + \delta J_{2,on}$, respectively. For gates, the combined activity of two output promoters in series $\delta J_1 + \delta J_2$ was calculated as $\delta J_{1,out} + \delta J_{2,out}$, where the output promoter activity of each gate ($\delta J_{1,out}$ and $\delta J_{2,out}$) was given by equation 4 with parameters $\delta J_{out}^{min}$, $\delta J_{out}^{max}$, *K*, and *n*. For each state, to calculate the expected output promoter activity of a gate, the total input RNAP flux $J_{in}$ was extracted from the transcription profile and used as input to equation 4 (Fig 1D). Parameters for sensors ($\delta J_{on}$ and $\delta J_{off}$) and gates ($\delta J_{out}^{min}$, $\delta J_{out}^{max}$, *K* and *n*) were then fitted to minimize the adapted error function. To extract the individual contributions of each promoter to the combined measured activity of the output promoters $\delta J_{1+2,i}$ (Fig 4B and Appendix Fig S7B), we calculated the fractional contribution and split the measured combined activity according to this. Specifically, the estimated measured activity of output promoter 1 for state *i* was calculated as $\delta J_{1+2,i} \times [\delta J_{1,i}/(\delta J_{1,I} + \delta J_{2,i})]$, and for promoter 2 as $\delta J_{1+2,i} \times [\delta J_{2,i}/(\delta J_{1,I} + \delta J_{2,i})]$.

### Calculation of the predicted transcription profiles

Predicted promoter activities in RPUs were calculated by Cello and converted to arbitrary units (au) compatible with the experimentally measured transcription profiles by using the conversion factor 1 RPU = 2,895 au. The circuit DNA is scanned from the 5′- to 3′-end to trace out the predicted profile. Upon reaching a promoter, the height of the profile is raised at the transcription start site by the predicted promoter strength (from Cello in au) multiplied by $(1-p_c)$, where $p_c$ is the cleavage efficiency (Nielsen *et al*, 2016) of the next downstream ribozyme. Upon reaching a ribozyme, the profile height it set to the combined strength of upstream promoters directly driving expression (from Cello in au) plus any read-through from the nearest upstream terminator. This height is traced along the circuit until the end of the next downstream terminator is reached. Then, to capture RNAP read-through, the profile height is multiplied by $1/T_s$, where $T_s$ is the terminator strength from Chen *et al* (2013).

### Measurement of doubling times

Individual colonies were inoculated into MOPS EZ Rich Defined Medium (Teknova, CA, M2105) with 0.2% glycerol carbon source and 50 μg/ml kanamycin (Gold Biotechnology, MO, K-120-5) and grown overnight for 16 h at 37°C and 1,000 rpm in V-bottom 96-well plates (Nunc, Roskilde, Denmark, 249952) in an ELMI Digital Thermos Microplates shaker incubator (Elmi Ltd, Riga, Latvia). The following day, cultures were diluted 178-fold (16.9 μl into 3 ml) into EZ Rich glycerol with kanamycin and grown in an Innova 44 shaker (Eppendorf, CT) at 37°C and 250 rpm for 3 h. Next, 1 ml of culture was added to a VWR disposable cuvette (VWR, PA, 97000-586) and the optical density at 600 nm (OD_{600}) was measured using a Cary 50 Bio spectrophotometer (Agilent, CA, 10068900). For culture tube assays (Falcon 14-ml round-bottom polypropylene tubes; Corning, MA, 352059), cells were diluted 658-fold (4.56 μl into 3 ml) into EZ Rich glycerol with kanamycin and inducers. For Erlenmeyer flask assays (Pyrex 250 ml; Cole-Palmer, IL, 4980-250), cells were diluted 658-fold (76 μl into 50 ml) into EZ Rich glycerol with kanamycin and

inducers. To calculate the initial $OD_{600}$ post-dilution, the aforementioned $OD_{600}$ value was divided by 658. Culture tubes and Erlenmeyer flasks were then grown in an Innova 44 shaker at 37°C and 250 rpm for five hours. The final $OD_{600}$ for each vessel/inducer condition was measured for 1 ml of culture in the manner described above. The doubling time was calculated as $(5 \text{ h})/\log_2(OD_{Final}/OD_{Initial})$.

### Characterization of the modified circuit

New genetic parts were synthesized as gBlock gene fragments (Integrated DNA Technologies, CA) and assembled into full genetic circuit using Gibson assembly (New England Biolabs, MA, 2611L). The modified circuit was sequence verified and transformed into *E. coli* DH10-beta (New England Biolabs, MA, C3019). Cultures were grown in culture tubes under different induction conditions, and RNA-seq samples were prepared by following the protocols described above.

### Data availability

RNA-seq data collected in this study were deposited to Gene Expression Omnibus (https://www.ncbi.nlm.nih.gov/geo/) under the accession numbers GSE88835 and GSE98890. Python scripts that implement the characterization pipeline are released as open-source software under the MIT license (Computer Code EV1; GitHub repository: https://github.com/VoigtLab/MIT-BroadFoundry).

Expanded View for this article is available online.

### Acknowledgements

This work was supported by US Defense Advanced Research Projects Agency (DARPA) Living Foundries awards HR0011-13-1-0001, HR0011-12-C-0067 and HR0011-15-C-0084 (C.A.V., T.E.G. and D.B.G.); US Department of Commerce − NIST (National Institute of Standards and Technology) award 70NANB16H164 (C.A.V.); Office of Naval Research, Multidisciplinary University Research Initiative grant N00014-13-1-0074 (C.A.V. and A.A.K.N.); Samsung Scholarship (Y.P.).

### Author contributions

CAV, TEG, YP, and AAKN conceived of the study. CAV, TEG, AEB, and YP wrote the manuscript. TEG, AEB, BSD, and DBG developed algorithms and performed simulations. AAKN, YP, and JZ carried out the experiments.

### Conflict of interest

The authors declare that they have no conflict of interest.

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
