## [Review Process File · Molecular Systems Biology]

Genetic circuit characterization and debugging using RNA-seq

Thomas E. Goroehowski, Amin Espah Borujeni, Yongjin Park, Alec A.K. Nielsen, Jing Zhang, Bryan S. Der, D. Benjamin Gordon & Christopher Voigt

Corresponding author: Christopher Voigt, Massachusetts Institute of Technology and Broad Institute of MIT & Harvard

Review timeline:

Submission date:	24 November 2016
Editorial Decision:	05 January 2017
Revision received:	25 August 2017
Editorial Decision:	25 September 2017
Revision received:	25 September 2017
Accepted:	28 September 2017

Editor: Maria Polychronidou

Transaction Report:

1st Editorial Decision

05 January 2017

Thank you for submitting your work to Molecular Systems Biology. We have now heard back from the two referees who agreed to evaluate your study. As you will see below, both reviewers appreciate that applying RNA-seq to the characterization of synthetic circuits is an interesting approach. However, they list several concerns, which we would ask you to address in a revision.

I think that the reviewers' recommendations are rather clear so there is no need to repeat the points listed below, but please let me know in case you would like to discuss any specific point. As both reviewers point out, further analyses illustrating how the presented approach can be applied for circuit debugging would significantly enhance the impact of the study.

REVIEWER REPORTS

Reviewer #1:

Summary.

In the manuscript under consideration Goroehowski et al. develop a strategy to characterise and quantify the functionality of synthetic gene circuit based on logic gates in *E. coli* using RNA

sequencing (RNAseq). The paper describes the development and application of an RNAseq experimental and analysis pipeline which is applied to the characterization of a model 3-input 1-output combinatorial logic circuit. The genetic circuit was considered in different functional states, with samples collected when different permutations of input signals were used. RNAtag-seq was used for library preparation, raw reads were converted to transcriptional profiles and then genetic parts and circuit characterisation were performed with the aid of mathematical modelling. Promoter and terminator strengths were calculated based on transcriptional analysis and compared to values known from prior literature and from fluorescent protein reporter expression measurements. Most of the promoters and terminators performed as expected with some interesting exceptions. To examine this further, the genetic sensors composing the circuit were also characterised in isolation and also when within the circuit in order to identify context dependencies. Strong context effects were observed for a few promoters and logic gates, highlighting otherwise-hidden failure modes. Alongside the performance of the model circuit, the RNAseq pipeline also examined its impact on the host cell. The expression profile of the host was analysed in different growth conditions while running the model circuit, which led to the identification of clusters of native genes that are upregulated and downregulated as the circuit is functioning. Finally, the authors also identified antisense transcriptions occurring in their circuit due to the presence of cryptic promoters on the antisense strand of the constructs. Their analyses highlighted the impact of these antisense transcriptions on the termination efficiency of some of the terminators used.

General Remarks.

The work presented in the manuscript addresses a significant challenge in synthetic biology: the characterisation of genetic parts within the context of complex regulatory circuits. This greatly enables the identification of failure mechanisms for genetic components. The work also aims to overcome the limitations of using fluorescent reporters as the sole method of obtaining characterization data, and instead proposes that RNAseq can offer a more direct measurement of parts functionality. With their analysis the authors clearly show that RNAseq can represent a powerful tool for the quantification of parts behavior in the context of transcription-based regulatory circuits. Overall I suggest that this paper is suitable for publication in your journal once the below points are addressed.

Major points

While the work presented is impressive, I am not completely convinced that the proposed strategy can represent a widely-utilisable method for the community (as stated is by the authors). First of all, RNAseq is still a very expensive procedure for most labs, even if the advent of newly developed methods such as the RNAtag-seq is now opening the path to decreasing costs.

Secondly, even if the authors show that RNAseq combined with their newly automated pipeline can guide the quantification and characterisation of parts behaviour in the context of a complex genetic system, they do not suggest or show how this can then in turn help the design of systems where the identified failures can be avoided or overcome. For example, it would greatly help advance the field if the RNAseq analysis pipeline was also paired with a robust fault debug workflow. Therefore in consideration of these points, can the authors show how their system could be used to design genetic circuits with optimised behaviour? Without this extra step the research proposed represents elegant work but it does not help towards wider debugging of genetic circuits and the improvement of their designs.

Minor points

A. Page 4 line 7: I would ask the authors to explain this better. Indeed, the real novelty of the work by Shishkin et al. is the tagging of total RNA fragments preceding ribodepletion rather than just the tagging of the different pooled samples, which was already an established protocol.

B. Page 4 line 10: can authors clarify the sentence? One single Illumina HiSeq 2500 flow cell should give around 250×10^6 reads. Pooling 1000 samples would give 3×10^5 reads per sample, not enough to run a differential gene expression analysis among samples in terms of their genomic transcriptional profile [also accordingly to Haas et al. BMC Genomics 2012, 13:734]. This does not really make RNAseq a scalable technique considering the costs that still this would require for a very high number of samples.

C. page 9 line 19: "Due to the of the use of" should be replaced by "Due to the use of"

D. page 26 Figure 1 legend: I would suggest authors to write a more self-explanatory legend for Figure 1 describing the meaning of symbols and panels.

Reviewer #2:

Gorochowski et al. describe the application of RNA-Seq to characterize the performance of a synthetic biological circuit. Systems biologists often use -omics tools, such as RNA-seq, to analyze the change in gene expression profiles in response to growth conditions, stress, etc. This manuscript describes the first application of this transcriptomics approach to the characterization of a synthetic genetic circuit. As circuits grow in complexity and size, characterizing their performance and identifying failure points becomes increasingly challenging, as circuit characterization is often performed with a limited repertoire of fluorescent proteins that serve as outputs. End-point characterization of an output prohibits a more holistic assessment of circuit performance across a number of states and individual parts. Gorochowski et al. are the first to apply the well-established RNA-Seq technology to simultaneously characterize all the components of their large synthetic circuit and to analyze the effect of a synthetic genetic circuit on the transcriptome of the host organism. The major advance is conceptual in nature in applying existing methods to characterizing synthetic circuits, though there are additional advances presented in the development of models and algorithms specific for analysis of synthetic circuits (i.e. transcription profiles), rather than those used by systems biologists.

The circuit investigated here involved 3 inputs (aTc, IPTG, Arabinose) and a single output (YFP). A total of 8 states are possible with the circuit design and each combinatorial possibility is analyzed by RNAtag-seq, which allows for multiplexed analysis of the various states in a single high-throughput sequencing run. To perform these experiments, *E. coli* DH10B cells harboring the genetic circuit were cultured in 2 different conditions for 8 states, representing various combinations of inputs. The cells were then flash frozen and RNA was recovered and ligated for conversion to cDNA. Libraries corresponding to the different conditions were tagged with sequencing barcodes to multiplex for sequencing. Raw RNA-seq reads were subject to a suite of bioinformatics tools to convert the raw data into transcription profiles that attempt to correct for biases introduced by the upstream steps. Genetic parts are characterized by the development of several models.

After investigating the circuit under the 8 states of 3 inputs, the authors explore the same 8 states under two different growth conditions (tubes vs. flasks). This transcriptomic approach to circuit characterization is claimed to have allowed the authors to assess the individual response functions of the logic gates, analyze the impact of host gene expression and identify failure points, which would otherwise be obscured by simple measurement of a single reporter protein.

Overall this is a very important topic of work - RNA-Seq is a powerful tool that has been around for quite a while and it is a bit surprising that it has not been used as a tool in synthetic biology before. Since many genetic circuits involve passing signals at the transcriptional level at some point, this offers a potentially profoundly powerful tool for assessing the internal working of genetic circuits as the authors demonstrate. In addition, the potential to account for off target effects by sequencing the entire transcriptome is a huge advantage. The key conclusion that can be drawn from this work is that RNA-Seq is a powerful tool to analyze synthetic biological circuits. This conclusion is backed up by the data suggesting several problematic areas for debugging.

The topic of the manuscript is exciting, and we imagine it having wide impact among synthetic biologists. However we have several major and minor points that should be addressed prior to publication.

Major Points:

- 1.) Overall the authors presented convincing evidence that their measurements correlate well with the behavior of the genetic circuit tested. However, a number of new analyses were developed in this work (for example re-normalizing RNA-Seq distribution across a transcript, and taking differences in these distributions across part boundaries to calculate flux of RNA polymerase) and while these make sense, they were not experimentally validated properly to be able to assess the accuracy of the quantitative predictions of the method.

In order to do this, basic validation experiments should be performed in order to be able to fully establish the accuracy of their new data analysis developments. While the authors place great care in describing their new data analysis methods (such as re-normalizing RNA-Seq distribution across a transcript, and taking differences in these distributions across part boundaries to calculate flux of RNA polymerase) in both Figure 1 and the supplementary information, they jump right into using them in the context of a complicated model circuit. The authors should attempt to perform experiments on single parts - much like the schematic examples in Figure 1 - in order to validate their data analysis on simpler systems.

2) The authors claim that RNA-seq is a "powerful method for circuit debugging" yet they make no attempt at debugging. Rather, they utilize RNA-Seq as a method to identify the bugs. While identification of bugs alone is an important first step, the work would be strengthened by at least attempting to rectify the identified failure modes. For example - does the observed failure mode suggest a fix? Showing that circuits can be fixed using suggestions from their data analysis would greatly strengthen the overall approach.

3) No replicates were performed for the various states. The "technical replicates" described are sequencing replicates. Replicate experiments should be performed and analyzed.

4) The manuscript reads like a technical manual, particularly in the beginning. While this writing style would be suitable for a methods development article, it seems out of place for an original research article presenting novel biological insight. The manuscript also lacks in some of the motivation for doing various manipulations - it would help to justify why they made certain choices rather than stating the choices made.

Minor points:

1) The first paragraph of the data collection section of the results mentions "high RNA integrity numbers (RIN > 8.5)" but fails to provide a definition or any context for the reader to interpret what constitutes high vs. low RIN.

2) Under flow cytometric analysis in the methods, the number of events counted is in the range of 10^3 to 10^5 . 10^5 is usually the standard number of events collected during flow analysis. The authors should address this discrepancy and provide the number of events per sample.

3) The authors should justify their use of lesser known bioinformatics tools over the gold standard Tuxedo software suite (Cufflinks, Bowtie, etc.).

4) The authors should also justify their motivation for studying the circuit under investigation. It's not clear why this particular circuit was chosen, apart from a sentence stating the circuit was chosen as a "model system." Model systems in biology typically interest a broad range of scientists studying the same model system. An example of a more appropriate model system for synthetic biologists could be the violacein pathway. Any circuit which has been independently investigated by labs other than the authors would be more suitable.

5) On p.6 the authors state a formula for the flux of RNA polymerase as a function of the RNA degradation rate and the measured (and re-normalized) transcript profile $M(x)$. Unlike other parts of the manuscript, this formula is left unjustified. Presumably it comes from writing a model of transcription flux along a transcript that looks something like: $dM(x)/dt = J(x-1,t) - \gamma M(x,t)$, though at steady-state this gives a slightly different result from what is stated in the text. The authors should provide further justification for this formula since it is central to their data analysis approach.

6) On p. 8, the authors give a formula for J_{in} for a composite two promoter system in terms of δJ 's from the individual promoters. However, in this case wouldn't the flux simply be just the flux from the 3'-most promoter junction since all RNAP would have to go through that junction? The current formula seems to over-estimate the flux from the composite promoter. This would be a great place for simpler validation experiments to validate this formula since it is core to the analysis presented.

- 7) The third paragraph of the introduction states "Transcriptomic methods, such as RNA sequencing (RNA-seq) enable the measurement of genome-wide mRNA levels with base pair resolution." Unfortunately, base pairs are not yet being resolved genome-wide with RNA-Seq technologies. It should say nucleotide (or nucleobase) resolution, rather than base pair.
- 8) The sixth paragraph of the "applications to a model circuit" section of the results refers the ON/OFF activities of the reporters as being illustrated in black and red, but a figure is not referenced. I assume this is figure 3.
- 9) It would have been nice to have access to the code for reviewing purposes.
- 10) Figure 3 - the style of this plot was very confusing and it was hard to verify the discussion around these figures.
- 11) SI Page 3 - In general the derivations and example figures presented were very well done and much appreciated.
- 12) SI Page 3 - Please provide explicit formulas for the A₅ and A₃ profile landscapes.
- 13) SI Figure 3E - this looks like the unbiased correction profile plot - is that a mistake or correct?
- 14) It almost looks like the author's method over-corrects M(x) along the transcript boundaries (i.e. the profiles go up around the edges). Can the authors comment on this? How would an over-correction affect downstream analysis?

1st Revision - authors' response

25 August 2017

Reviewer #1:

1. *While the work presented is impressive, I am not completely convinced that the proposed strategy can represent a widely-utilisable method for the community (as stated is by the authors). First of all, RNAseq is still a very expensive procedure for most labs, even if the advent of newly developed methods such as the RNAtag-seq is now opening the path to decreasing costs. Secondly, even if the authors show that RNAseq combined with their newly automated pipeline can guide the quantification and characterisation of parts behaviour in the context of a complex genetic system, they do not suggest or show how this can then in turn help the design of systems where the identified failures can be avoided or overcome. For example, it would greatly help advance the field if the RNAseq analysis pipeline was also paired with a robust fault debug workflow. Therefore, in consideration of these points, can the authors show how their system could be used to design genetic circuits with optimised behaviour? Without this extra step, the research proposed represents elegant work but it does not help towards wider debugging of genetic circuits and the improvement of their designs.*

We have included new data showing the application of the RNA-seq data to debug a circuit by guiding part replacement to remove unwanted antisense transcription (Figure 5B and edits made to Results and Materials and Methods).

RNA-seq has become a common tool in biology and numerous computational tools have been published. The potential impact on synthetic biology is significant and the cost, which is dropping rapidly as sequencing costs decline, should not preclude publication. Most labs have access to facilities that can perform RNA-seq at a cost of several hundred dollars a sample.

2. *Page 4 line 7: I would ask the authors to explain this better. Indeed, the real novelty of the work by Shishkin et al. is the tagging of total RNA fragments preceding ribodepletion rather than just the tagging of the different pooled samples, which was already an established protocol.*

The sentence has been updated as suggested: "Recently, a method called RNAtag-seq (Shishkin et al, 2015) was developed that uses nucleotide barcodes to tag total fragmented RNA before depletion of ribosomal-RNA (rRNA) to allow for many samples to be efficiently pooled and sequenced together."

3. Page 4 line 10: can authors clarify the sentence? One single Illumina HiSeq 2500 flow cell should give around 250×10^6 reads. Pooling 1000 samples would give 3×10^5 reads per sample, not enough to run a differential gene expression analysis among samples in terms of their genomic transcriptional profile [also accordingly to Haas et al. BMC Genomics 2012, 13:734]. This does not really make RNAseq a scalable technique considering the costs that still this would require for a very high number of samples.

The back-of-the-envelope calculation in this comment is a little off. A single Illumina HiSeq 2500 flow cell consists of 8 lanes and in total produces ~4 billion paired-end reads per run. This would result in ~4 million paired-end reads per sample, which is sufficient for our characterization method. In the Haas *et al.* paper raised by the reviewer, it is stated that 5-10 million reads per sample are sufficient for most RNA-seq applications, but they also emphasize that a significantly greater multiplexing of samples within a run will lead to only modest drops in sensitivity for many applications (e.g., differential expression of genes). Therefore, our statement of “up to 1000 samples” being able to be characterized on a single HiSeq 2500 run is realistic. We have updated the sentence to include this information and have included a citation to the paper mentioned by the reviewer: “This approach can be scaled-up: a single flow cell on an Illumina HiSeq 2500 machine generates ~4 billion paired-end reads and is therefore capable of characterizing up to 1000 samples (Haas et al, 2012), which could be used to simultaneously assay many different circuits and states.”

4. Page 9 line 19: "Due to the of the use of" should be replaced by "Due to the use of".

We have updated this sentence as suggested.

5. Page 26 Figure 1 legend: I would suggest authors to write a more self-explanatory legend for Figure 1 describing the meaning of symbols and panels.

The caption for Figure 1 has been expanded to more clearly describe the panels and symbols used.

Reviewer #2:

1. Overall the authors presented convincing evidence that their measurements correlate well with the behavior of the genetic circuit tested. However, a number of new analyses were developed in this work (for example re-normalizing RNA-Seq distribution across a transcript, and taking differences in these distributions across part boundaries to calculate flux of RNA polymerase) and while these make sense, they were not experimentally validated properly to be able to assess the accuracy of the quantitative predictions of the method. In order to do this, basic validation experiments should be performed in order to be able to fully establish the accuracy of their new data analysis methods. While the authors place great care in describing their new data analysis methods (such as re-normalizing RNA-Seq distribution across a transcript, and taking differences in these distributions across part boundaries to calculate flux of RNA polymerase) in both Figure 1 and the supplementary information, they jump right into using them in the context of a complicated model circuit. The authors should attempt to perform experiments on single parts - much like the schematic examples in Figure 1 - in order to validate their data analysis on simpler systems.

We have included new data using three biological replicates to help validate the model in terms of its ability to account for day-to-day variation in the culture conditions and sequencing prep (e.g., changes in the fragment distribution). These data now appear extensively in the paper. Figure 1 contains averaged profiles and the variation is shown in the SI. Part and device quantification in the tables now have error bars.

We selected not to validate the methods on individual parts. The purpose of the software is to provide a means to convert an observed profile into part strengths – not standardize part measurements – and we develop simple models to do accomplish this task. The part strengths (e.g., fluxes) are still presented in arbitrary units, so it is not clear how repeating the

experiments on an individual well-characterized promoter would help or impact our claims. In the long run, it would be great to standardize a promoter using RNA-seq and biophysical methods to be able to convert the arbitrary units to actual fluxes, but this is well outside the scope of this work.

2. *The authors claim that RNA-seq is a "powerful method for circuit debugging" yet they make no attempt at debugging. Rather, they utilize RNA-Seq as a method to identify the bugs. While identification of bugs alone is an important first step, the work would be strengthened by at least attempting to rectify the identified failure modes. For example - does the observed failure mode suggest a fix? Showing that circuits can be fixed using suggestions from their data analysis would greatly strengthen the overall approach.*

To demonstrate how our analysis can be used for debugging, our results were used to improve circuit function by removing unwanted antisense transcription through guided part replacement (Figure 5B and edits made to Results and Materials and Methods). New RNA-seq data show that this change disrupts antisense transcription as it was designed to do.

3. *No replicates were performed for the various states. The "technical replicates" described are sequencing replicates. Replicate experiments should be performed and analyzed.*

Three biological replicates of RNA-seq experiments have been performed for the circuit across all input states for cells grown in culture tubes. These data show high-reproducibility (Figure S2) and has been incorporated into the updated manuscript (edits made to Results and Methods, Figures 2, S2 and S3, and Tables 1, 2 and S3).

4. *The manuscript reads like a technical manual, particularly in the beginning. While this writing style would be suitable for a methods development article, it seems out of place for an original research article presenting novel biological insight. The manuscript also lacks in some of the motivation for doing various manipulations - it would help to justify why they made certain choices rather than stating the choices made.*

We have edited the manuscript to address this point and have added insight behind the choices made.

5. *The first paragraph of the data collection section of the results mentions "high RNA integrity numbers (RIN > 8.5)" but fails to provide a definition or any context for the reader to interpret what constitutes high vs. low RIN.*

To ensure the reader is aware of what constitutes a high and low RIN number, we added new sentences in the Methods summarizing this information with a reference to the Schroeder *et al.* (2006) paper that describes the methodology in detail: "We also check the quality of the RNA extracted by calculating the RNA integrity number (RIN), which ranges from a value of 10 if all RNA is intact, to 1 if the RNA is totally degraded (Schroeder *et al.*, 2006). We only consider highly intact RNA samples with a RIN > 8.5 (Imbeaud *et al.*, 2005)." Note that this sentence and interpretation got moved out of the results and into the methods to reduce it sounding like a technical manual (point #4).

6. *Under flow cytometric analysis in the methods, the number of events counted is in the range of 10^3 to 10^5 . 10^5 is usually the standard number of events collected during flow analysis. The authors should address this discrepancy and provide the number of events per sample.*

The cytometry has been re-run such that the number of counts is consistently >20,000, which is typical.

7. *The authors should justify their use of lesser known bioinformatics tools over the gold standard Tuxedo software suite (Cufflinks, Bowtie, etc.).*

For read mapping, BWA was chosen because it is proven to be sensitive and accurate when using its default parameters (Hatem *et al.* *BMC Bioinformatics* 14:184, 2013). Unlike BWA, Bowtie does not support indels, which can significantly reduce the reads mapped to a

reference, and while similar sensitivity can be achieved with Bowtie2, this often involves careful and laborious tuning of parameters (Hatem *et al. BMC Bioinformatics* 14:184, 2013). For sample normalization and differential gene expression, edgeR was chosen because it has been shown to accurately calculate differentially expressed genes and can tolerate unbalanced library sizes and low sequencing depths (a potential issue with highly multiplexed sequencing libraries) better than other tools, e.g. Cuffdiff2 and DESeq (Zhang *et al. PLoS ONE* 9:e103207, 2014).

8. *The authors should also justify their motivation for studying the circuit under investigation. It's not clear why this particular circuit was chosen, apart from a sentence stating the circuit was chosen as a "model system." Model systems in biology typically interest a broad range of scientists studying the same model system. An example of a more appropriate model system for synthetic biologists could be the violacein pathway. Any circuit which has been independently investigated by labs other than the authors would be more suitable.*

We have removed the reference to this circuit being a “model system.” We selected this circuit because it is representative of class of combinatorial logic circuits that we are particularly interested in – and are common in the field – and we knew that it failed when the growth conditions were changed. Applying the software to a metabolic pathway would be outside the scope of the manuscript.

9. *On p.6 the authors state a formula for the flux of RNA polymerase as a function of the RNA degradation rate and the measured (and re-normalized) transcript profile $M(x)$. Unlike other parts of the manuscript, this formula is left unjustified. Presumably it comes from writing a model of transcription flux along a transcript that looks something like: $dM(x)/dt = J(x-l, t) - \gamma M(x, t)$, though at steady-state this gives a slightly different result from what is stated in the text. The authors should provide further justification for this formula since it is central to their data analysis approach.*

We have edited the text to clarify the derivation.

10. *On p. 8, the authors give a formula for J_{in} for a composite two promoter system in terms of δJ 's from the individual promoters. However, in this case wouldn't the flux simply be just the flux from the 3'-most promoter junction since all RNAP would have to go through that junction? The current formula seems to over-estimate the flux from the composite promoter. This would be a great place for simpler validation experiments to validate this formula since it is core to the analysis presented.*

Yes, J_{in} is the total flux at the 3'-most promoter junction and the δJ 's are how much each promoter in series contributes to this total flux.

11. *The third paragraph of the introduction states "Transcriptomic methods, such as RNA sequencing (RNA-seq) enable the measurement of genome-wide mRNA levels with base pair resolution." Unfortunately, base pairs are not yet being resolved genome-wide with RNA-Seq technologies. It should say nucleotide (or nucleobase) resolution, rather than base pair.*

We have changed the sentence as suggested.

12. *The sixth paragraph of the "applications to a model circuit" section of the results refers the ON/OFF activities of the reporters as being illustrated in black and red, but a figure is not referenced. I assume this is figure 3.*

We have changed the sentence as suggested.

13. *It would have been nice to have access to the code for reviewing purposes.*

The code will be made publically available via GitHub (noted in the Data Availability section).

14. *Figure 3 - the style of this plot was very confusing and it was hard to verify the discussion around these figures.*

The figure has been updated to show a smaller area around each promoter and terminator part. This makes it easier to see the individual lines corresponding to each transcription profile and the major changes in their levels. The similar Appendix Figure S6 has been updated in the same way.

15. *SI Page 3 - In general the derivations and example figures presented were very well done and much appreciated.*

Thank-you.

16. *SI Page 3 - Please provide explicit formulas for the A_5 and A_3 profile landscapes.*

All terms are now explicitly defined in either Box 1 or Appendix Text S1.

17. *SI Figure 3E - this looks like the unbiased correction profile plot - is that a mistake or correct?*

Note that this Figure has been edited to reflect the improved method.

18. *It almost looks like the author's method over-corrects $M(x)$ along the transcript boundaries (i.e. the profiles go up around the edges). Can the authors comment on this? How would an over-correction affect downstream analysis?*

When describing our improved correction method in Appendix Text S1, we show that for most transcription units the sequenced fragments map uniformly at random along their length (Appendix Figure S1). This ensures effective correction using our method.

The correction method is described in Appendix Text S1. Most transcription units do not show this overcorrection. When it does happen, it is due to biases in the location of the mapped fragments. There are two cases where this occurs: state $-/+$ for the *LitR* and *BM3R1* transcription units. For these, there is an enrichment of reads mapping to the 5'-end of each transcription unit (Appendix Figure S1B), which causes an increase in the profile at these points. When using the data to calculate a part strength, a window is applied to remove the impact of these localized effects.

2nd Editorial Decision

25 September 2017

Thank you for sending us your revised manuscript. We have now heard back from the referee who was asked to evaluate your study. As you will see below, s/he is satisfied with the modifications made and thinks that the study is now suitable for publication. In line with the recommendation of the reviewer, we would ask you to provide the code used for data analysis (i.e. if there are additional files that have not been deposited in GitHub).

 REVIEWER REPORT

Reviewer #2:

The authors have done a thorough job of addressing most of our major points and all of our minor points. The only point to emphasize is the authors should release their data analysis code to make sure this technique can be used by the field. With the addition of new experiments, and once the code is released, we now feel this is suitable for publication.

2nd Revision - authors' response

25 September 2017

Reviewer #2:

The authors have done a thorough job of addressing most of our major points and all of our minor points. The only point to emphasize is the authors should release their data analysis code to make sure this technique can be used by the field. With the addition of new experiments, and once the code is released, we now feel this is suitable for publication.

We have included a .zip file of the code used in the study (Computer Code EV1) and updated the GitHub repository link in the Data Availability section.

Corresponding Author Name: Christopher A. Voigt

Manuscript Number: MSB-16-7461